



# Ozone trends in homogenized Umkehr, Ozonesonde, and COH overpass records

Irina Petropavlovskikh[1,2], Jeannette D. Wild[3,4], Kari Abromitis[1,2], Peter Effertz[1,2], Koji Miyagawa[5], Lawrence E. Flynn[4], Eliane Maillard-Barras[6], Robert Damadeo[7], Glen McConville[1,2], Bryan Johnson[2], Patrick Cullis[2], Sophie Godin-Beckmann[8], Gerard Ancellet[8], Richard Querel[9], Roeland Van Malderen[10], Daniel Zawada[11]

[1] CIRES, University of Colorado, Boulder, CO, USA
[2] NOAA, Global Monitoring Lab, Boulder, CO, USA
[3] University of Maryland, Earth System Science Interdisciplinary Center (ESSIC), College Park, MD, USA
[4] NOAA/NESDIS/Center for Satellite Applications and Research (STAR), College Park, MD, USA
[5] NEIS, Tsukuba-City, Japan
[6] Federal Office of Meteorology and Climatology, MeteoSwiss, Payern, Switzerland
[7] NASA Langley Research Center, Hampton, VA, USA
[8] LATMOS Sorbonne Université, UVSQ-CNRS/INSU, Paris, France.
[9] National Institute of Water & Atmospheric Research (NIWA), Lauder, New Zealand
[10] Royal Meteorological Institute of Belgium, Uccle (Brussels), Belgium
[11] University of Saskatchewan, Saskatoon, SK, Canada

*Correspondence to:* Irina Petropavlovskikh (Irina.petro@Noaa.gov)

**Abstract.** This study presents an updated evaluation of stratospheric ozone profile trends at Arosa/Davos/Hohenpeißenberg, Switzerland/Germany, Observatory de Haute Provence (OHP), France, Boulder, Colorado, Mauna Loa Observatory (MLO) and Hilo, Hawaii, and Lauder, New Zealand with focus on the ozone recovery period post 2000. Trends are derived using vertical ozone profiles from NOAA's Dobson Network via the Umkehr method (with a recent new homogenization), ozonesondes, and the NOAA COHesive SBUV/OMPS satellite-based record (COH) sampled to match geographical coordinates of the ground-based stations used in this study. Analyses of long-term changes in stratospheric ozone time series were performed using the updated version (0.8.0) of the Long-term Ozone Trends and Uncertainties in the Stratosphere (LOTUS) Independent Linear Trend (ILT) regression model. This study finds a consistency of the trends derived from the different observational records, which is a key factor to the understanding of the recovery of the ozone layer after the implementation of the Montreal Protocol and its amendments that control ozone-depleting substances production and release into the atmosphere. The Northern Hemispheric Umkehr records of Aros/Davos, OHP, and MLO all show positive trends in the mid to upper stratosphere with trends peaking at ~+2%/decade. Although the upper stratospheric ozone trends derived from COH satellite records are more positive than those detected by the Umkehr system, the agreement is within the two sigma uncertainty. Umkehr trends in the upper stratosphere at Boulder and Lauder are positive but not statistically significant, while COH trends are larger and statistically significant (within 2 sigma). In the lower stratosphere, trends derived from Umkehr and ozonesonde records are mostly negative (except for positive ozonesonde trends at OHP), however the uncertainties are quite large. Additional dynamical proxies were investigated in the LOTUS model at five ground-based sites. The use of additional proxies did not significantly change trends, but equivalent latitude reduced the uncertainty of the Umkehr and COH trends in the upper stratosphere and at higher latitudes. In lower layers, additional predictors (tropopause pressure for all stations, two extra components of Quasi-Biennial Oscillation at MLO, Arctic



Oscillation at Arosa/Davos, OHP and MLO) improve the model fit and reduce trend uncertainties as seen by Umkehr
and sonde.

## 1 Introduction

The WMO Ozone Assessments (WMO, 2018; WMO, 2022), indicate that for some geographical regions, the
stratospheric ozone layer is recovering in accordance with the reduction of ozone depleting substances (ODS) whose
production was restricted by the Montreal Protocol and its amendments. The US Clean Air Act requires NOAA to
monitor prohibited chemicals and the ozone layer to ensure the success of the Montreal Protocol. NOAA's long-term
network of measurements helps to interpret total column and vertically resolved ozone changes and link ozone
recovery to the reduction of ODS levels in the stratosphere, changes in the lower stratosphere that are associated with
climate changes, and to the increases in the troposphere that are influenced by the stratosphere/troposphere exchange
and long-range transported pollution. The ongoing recovery of the stratospheric ozone layer is of great importance to
human health (i.e. cancer from enhanced UV exposures, Madronich et al., 2021), the sustained production of crops,
and the success of fisheries (dangerous algae blooms). For more information see the Environmental Effects
Assessment Panel 2022 Quadrennial Assessment (EEAP, 2023).
Studies of ozone recovery require long-term datasets often consisting of data merged from several instruments, or
from a single instrument type on multiple satellite platforms, or at a ground-based (GB) station. 2011 saw the initiation
of the SPARC/IO3C/IGACO-O3/NDACC (SI2N) activity to evaluate ozone trends in the depletion and recovery
phases using both GB and merged satellite observations. The resulting report (Harris et al., 2015) emerged near the
release of the 2014 WMO Ozone Assessment (WMO, 2014). The two studies resulted in broadly similar trend values
in both the depletion and recovery phases. But the WMO report determined the recovery trends to be statistically
significant, whereas the SI2N study did not. The discrepancy centered on differing error analysis techniques for the
merged datasets: SI2N using distribution of the individual variances around the mean and WMO using weighted mean
of the individual standard deviations.
The Long-term Ozone Trends and Uncertainties in the Stratosphere (LOTUS) study sought to reconcile the differences
between the WMO Assessment and the SI2N study. Phase 1 focused on developing best practices for data merging,
trend determination and error analyses. Results focused on analysis of broad latitudinal regions, near global, Northern
and Southern Hemisphere, and Tropics as were used in the SI2N studies. Results are found in the 2019 report
(SPARC/IO3C/GAW, 2019). Phase 2 refined the trend models, and extended the study to gridded, and GB ozone data
sets. The development of methods used in trend detection is built on the community knowledge gained during the
Tiger Team project in early 1990s (Reinsel et al., 2005), collaborations through the SPARC, WMO and IO3C
supported LOTUS activity (Hassler et al., 2014; Harris et al., 2015; Godin-Beekmann et al., 2022) and the most recent
contributions to the WMO Ozone assessment analyses published in Chapter 3, "Update on Global Ozone: Past, Present
and Future" (Hassler et al., 2022).
Understanding the causes of the differences between GB and satellite records can create improvements not only in the
internal consistency of data sets, but also in the uncertainties of overall ozone trends. Further, development of
techniques to directly assess uncertainties in the merged records resulting from discrepancies that cannot be completely



reconciled, such as small relative drifts and differences resulting from coordinate transformations and sampling
differences, allows for a more precise estimate of significance of the mean trends. For the GB and satellite data used
in the 2019 LOTUS Report, information on stability and drifts of the measurement was incomplete. The
homogenization of many ozonesonde records was recently addressed and data were reprocessed (Tarasick et al., 2016;
Van Malderen et al., 2016; Witte et al., 2017; Sterling et al., 2018; Witte et al., 2018; Ancellet et al., 2022) while some
instrumental artifacts still need to be addressed (Smit, 2021).
The first attempt to evaluate representativeness of the trends derived from GB station records in the middle and upper
stratosphere using SBUV data was done as a part of the LOTUS activity and was discussed in the 2019 LOTUS report.
Comparisons of trends derived from satellite data sub-sampled at the station location (overpass) to those derived from
the relevant zonal average provide a measure of potential sampling errors when comparing satellite and GB trends
(Zerefos et al., 2018; Godin-Beekmann et al., 2022). This paper continues that work by comparing trends derived from
several GB and satellite records that are matched spatially. We further investigate the impact of temporal matching on
trends.
The common statistical linear regression trend model used in the 2019 LOTUS Report and the 2022 update (Godin-
Beekmann et al., 2022) was optimized for analyses of the zonally averaged satellite data sets. However, analyses of
the GB and satellite overpass ozone profile data may require reconsideration of additional proxies and optimization
methods to improve interpretation of the processes that impact ozone changes over limited geophysical regions and
reduce trend uncertainties. An assessment of model sensitivities to uncertainties in the volcanic aerosols, solar cycle,
QBO, El Nino Southern Oscillation (ENSO) and other mechanisms also need to be considered in the GB and satellite
overpass record trend analysis. The localized time series for the assessment of dynamical and chemical proxies can
improve attribution of ozone variability, especially in the lower stratosphere, thus reducing uncertainties in the derived
trends. This paper provides an assessment of uncertainties in the derived trends from the NOAA ground-based,
ozonesonde and SBUV/OMPS (zonally averaged and overpass) records and reports improvements in the Multiple
Linear Regression (MLR) trend uncertainties with addition of proxies representing interannual dynamical variability
or long-term changes in atmospheric circulation.
In the LOTUS report, the ozone trends were analyzed at low and middle latitudes, with a focus on the upper and
middle stratosphere. This paper includes middle and low latitude trends assessed in the lower stratosphere and thus
offers an opportunity to test the additional proxy of the tropopause pressure (Thompson et al., 2021).
**2 Data**
**2.1 Umkehr and Ozonesonde Records at NOAA**
The Dobson Ozone Spectrophotometer has been used to study total ozone since its development in the 1920s
(Staehelin et al., 2018). Dobson records are regularly used in satellite record validation (Bai et al., 2015; Koukouli et
al., 2016; Boynard et al., 2018) and the development of global combined ozone data records (Fioletov et al., 2008;
Hassler et al., 2018). The NOAA Dobson ozone record was homogenized in 2017 to account for inconsistencies in
past calibration records, data processing methods and selection of representative data (Evans et al., 2017). NOAA



Dobson instruments at 4 stations and MeteoSwiss at Arosa/Davos also measure Umkehr profiles, which are derived
as partial column ozone amounts in ~5 km layers. Profiles are derived using an optimum statistical inversion of Dobson
measurements taken continuously at different solar zenith angles (SZAs) (Petropavlovskikh, 2005; Hassler, 2014).
These Umkehr data were recently homogenized to assure the removal of small but significant instrumental artifacts
that can impact the accurate detection of stratospheric ozone trends (Petropavlovskikh et al., 2022, Maillard Barras et
al., 2022). This study focuses on Umkehr records from the MeteoSwiss station of Arosa/Davos, Switzerland, and on
Umkehr records from the NOAA stations of Boulder, Colorado), Mauna Loa Observatory (MLO), Hawaii, Lauder,
New Zealand, and the Umkehr record from Observatory de Haute Provence (OHP), France. NOAA/GML for Umkehr
data means that the NOAA optimization process was applied to the operational records (N-values) prior to the retrieval
of ozone profiles. The source data used in this study are available at
https://gml.noaa.gov/aftp/data/ozwv/Dobson/AC4/Umkehr/Optimized/. See **Table 1** for details on the GB datasets,
locations, source of data and temporal extent of data used. Umkehr measurements are typically made twice per day
when there is no cloud obstruction.
The ozonesonde instrument has been flown at 4 NOAA stations since the 1980s. Evolving instrumentation and
standard operating procedures led to the development of data homogenization methods by NOAA and the international
community (i.e., ASOPOS-1, Smit, 2014) to resolve record inconsistencies in the NOAA (Sterling et al., 2018),
Canadian (Tarasick et al., 2016) and SHADOZ (Southern Hemisphere Additional Data from Ozonesonde) networks
(Witte et al., 2017; Witte et al., 2018). The effort was extended in the ASOPOS-2 (Smit et al., 2021) activity and
included a larger group of stations that are part of the NDACC (Network for Detection of Atmospheric Composition
Change) and WMO GAW (World Meteorological Organization Global Atmosphere Watch program) networks. The
error budget for each profile is calculated and included in the archived files (Sterling, 2018). Modern ozonesonde
instruments measure ozone at high vertical resolution, on the order of 100 m (Thompson et al., 2019) depending on
the altitude.
The sondes constitute an essential component of satellite calibration and cross-calibration (Hubert et al., 2016),
verification and improvement of climate chemistry and chemistry-transport models (Wargan et al., 2018; Stauffer et
al., 2019). The Dobson total ozone, Umkehr and ozonesonde profile records provide key measurements for upper and
middle stratospheric ozone trend calculations, and are part of the NOAA benchmark network for stratospheric ozone
profile observations (SPARC/IO3C/GAW, 2019; Godin-Beekmann et al., 2022; WMO, 2022).
The ozonesonde data are used for trend analyses from OHP, Boulder, and Lauder stations where we have Umkehr
observations.  Ozonesondes are launched at Hilo, Hawaii, which is nearly co-located with MLO.  Ozonesonde data
for the Arosa/Davos panel are selected from Hohenpeißenberg (HOH), Germany station that is in close vicinity to
Arosa/Davos station. Sonde measurements are typically measured once or twice per week, varying somewhat with
station operational procedures.
Data for the NOAA GML ozonesonde records are publicly available from the NOAA Global Monitoring Lab (GML)
at https://gml.noaa.gov/aftp/data/ozwv/Ozonesonde/. We use the '100 Meter Average Files' in each station directory.
Other sonde datasets used in this study are also available at several other data centers including the World Ozone and
Ultraviolet Radiation Data Centre (WOUDC, www.woudc.org), Network for the Detection of Atmospheric



Composition Change (NDACC, www.ndacc.org**)** data centers, or at the Harmonization and Evaluation of Ground-
based Instruments for Free-Tropospheric Ozone Measurements (HEGIFTOM, https://hegiftom.meteo.be/) archive.
Table 1 denotes the source of each dataset used in this study.
The ozonesonde data is of significantly higher vertical resolution (even when used as 100 m averages) than the Umkehr
data layers of approximately 5000 m. In order to create a dataset with comparable resolution, we use the Umkehr
averaging kernels (AK) to smooth the sonde data. Details appear in Appendix A. We cap the sonde profile at Umkehr
layer 5 (16–32 hPa) as there is not sufficient sonde information at higher altitudes to meet the requirements of the AKs
for layers 6 and above. We further match the ozonesonde data to the dates when both Umkehr and sonde data are
available using ± 24 hours to find a match, then generate the ozonesonde monthly mean. Appendix D explores the
impact of temporal sampling on trends. The final matched dataset, with AK averaging, is publicly available at
https://gml.noaa.gov/aftp/ozwv/Publications/2023_Umkehr_Ozone_Trends_Paper/.

| Location | WOUDC Station # | Instrument | Date Range used in trend calculations | Source |
|---|---|---|---|---|
| Arosa/Davos<br>Arosa, Switzerland (46.8° N, 9.7° E)<br>Davos, Switzerland (46.8° N, 9.8° E) | 035 | Umkehr | 1980 – 2018<br>2018 – 2020 | Optimization by NOAA/GML |
| Hohenpeißenberg (HOH), Germany<br>(47.8° N, 11.0° E) | 099 | Ozonesonde | 1980 – 2020 | NDACC |
| Observatory de Haute Provence (OHP), France  (43.9°N, 5.8° E) | 040 | Umkehr | 1983 – 2020 | NOAA/GML |
| | | Ozonesonde | 1991 – 2020 | NDACC (same as HEGIFTOM) |
| Boulder, Colorado<br>(40.0° N, 105.3° W) | 067 | Umkehr | 1980 – 2020 | NOAA/GML |
| | | Ozonesonde | 1980 – 2020 | NOAA/GML - 100m average data |
| Mauna Loa Observatory (MLO), Hawaii<br>(19.5°N, 155.6°W) | 031 | Umkehr | 1982 – 2020 | NOAA/GML |





| Hilo, Hawaii (19.7° N,155.1° W) | 109 | Ozonesonde | 1982 – 2020 | NOAA/GML - 100m average data |
| Lauder, New Zealand (45.0°S, 169.7°E) | 256 | Umkehr | 1987 – 2020 | NOAA/GML |
| | | Ozonesonde | 1987 – 2020 | NDACC |


**Table 1:  GB datasets, location, instrument type, temporal extent, data record source.  For the trend calculations we remove data during volcanic periods from 1982–1984 and 1991–1994.**

**2.2 The NOAA Cohesive (COH) Station Overpass Ozone Profile Datasets**

NASA and NOAA have produced satellite measurements of ozone profiles through the Solar Backscatter Ultraviolet (SBUV) on the sequence of Polar Orbiting Environmental Satellites (POES) since 1978.  This measurement series is extended with the related Ozone Mapping and Profiler Suite (OMPS) nadir profiler (NP) instruments using similar measurement techniques and retrieval algorithms.  These combine to provide nearly 45 years of continuous data (1978 – present). This single instrument type dataset eliminates many homogeneity issues including varying vertical resolution, or instrumentation differences.  Version 8.6 SBUV data incorporates additional calibration adjustments beyond the Version 8 release (McPeters et al., 2013).  Small but evident biases remain (Kramarova et al., 2013a). Several methods have been historically used to combine these datasets into a continuous series.  The NASA MOD version 1 dataset based on SBUV and OMPS v8.6 (Frith, 2014) combines data from all available satellites with no modification or bias adjustments.  NASA has developed an alternate processing for the SBUV and OMPS data (v8.7) which incorporates new calibrations at the radiance level, and updated a priori with improved troposphere. Additionally, the a priori is chosen to be representative of the local solar time of the measurement.  MOD v2 is based on the v8.7 processing (Frith et al., 2020), and further applies an adjustment to the v8.7 data to shift all measurements to a nominal measurement time of 1:30 PM local time.

The NOAA SBUV/2 and OMPS Cohesive dataset (further referred to as COH) combines data from the SBUV/2 and OMPS instruments using NASA's version 8.6 for the SBUV/2 data and NOAA/NESDIS version 4r1 for the OMPS Suomi National Polar-orbiting Partnership (SNPP) data.  This dataset uses correlation-based adjustments providing an overall bias adjustment plus an ozone-dependent factor (Wild et al., 2016) to moderate the remaining biases between instruments in the series. The resulting profile product is a set of daily or monthly zonal means and is publicly available at https://ftp.cpc.ncep.noaa.gov/SBUV_CDR. Zones are 5° wide in latitude, identified by the central latitude (2.5°, 7.5°, etc.).  Contributing satellites and their period of use is shown in Table 2.

| Satellite | Dates |
| --- | --- |
| Nimbus 07 | 10/1978 – 5/1989 |
| NOAA 11 | 6/1989 – 12/1993 |



| NOAA 09 | 1/1994 – 6/1997 |
|---------|-----------------|
| NOAA 11 | 7/1997 – 12/2000 |
| NOAA 16 | 1/2001 – 12/2003 |
| NOAA 17 | 1/2004 – 12/2005 |
| NOAA 18 | 1/2006 – 12/2010 |
| NOAA 19 | 1/2011 – 12/2013 |
| SNPP | 1/2014 – present |

**Table 2: Satellite mapping for COH data series**
A previous version of this dataset using OMPS v3r2 has been used in climate reviews and trend studies (Godin-
Beekmann et al., 2022; Weber et al., 2022a, 2022b) including Chapter 3 of the WMO Ozone Assessment (Hassler et
al., 2022). Appendix B examines the differences between the data versions. The impact on trends is limited to less
than 1% per decade, well within the precision of the trend results.
We create the overpass data at a ground station by collecting all profiles from a satellite within a ±2/20°
latitude/longitude box centred on the station. The box size is chosen to ensure that one to four points are found per
day. Fewer points are found if the orbit passes directly over the station; more points are found if the orbits straddle
the station. The collected profiles are inverse-distance weighted to the station location and averaged. COH style
adjustments are applied (Wild et al., 2016) creating a COH overpass time series from 1978 to present. This dataset is
available on the NOAA website at https://ftp.cpc.ncep.noaa.gov/SBUV_CDR/overpass.
Figure 1 shows the ozone anomaly time series for the 40–45° N zonal average data, and for the data at 3 stations in or
near the zone. Anomalies are calculated with respect to the zonal average climatology. The series shown are for the
layer data with the bottom pressure of the layer displayed on the right side of the graph. This depiction retains the
information about the relative differences between the stations and the zonal average. In the mid-stratosphere (25–10
hPa) the biases between the stations are most pronounced with Arosa/Davos usually showing less ozone and Boulder
usually showing more ozone. At the uppermost layers (1 and 4 hPa), and the lowest layer (41 hPa) the bias between
stations is reduced. The anomalies for Arosa/Davos and OHP, which are geographically closer than Boulder, are often
nearly anticorrelated with the Boulder anomalies especially in the second half of the year. Indeed at 16 hPa in
particular, one can see that often the Boulder anomalies are positive when the Arosa/Davos and OHP anomalies are
negative.

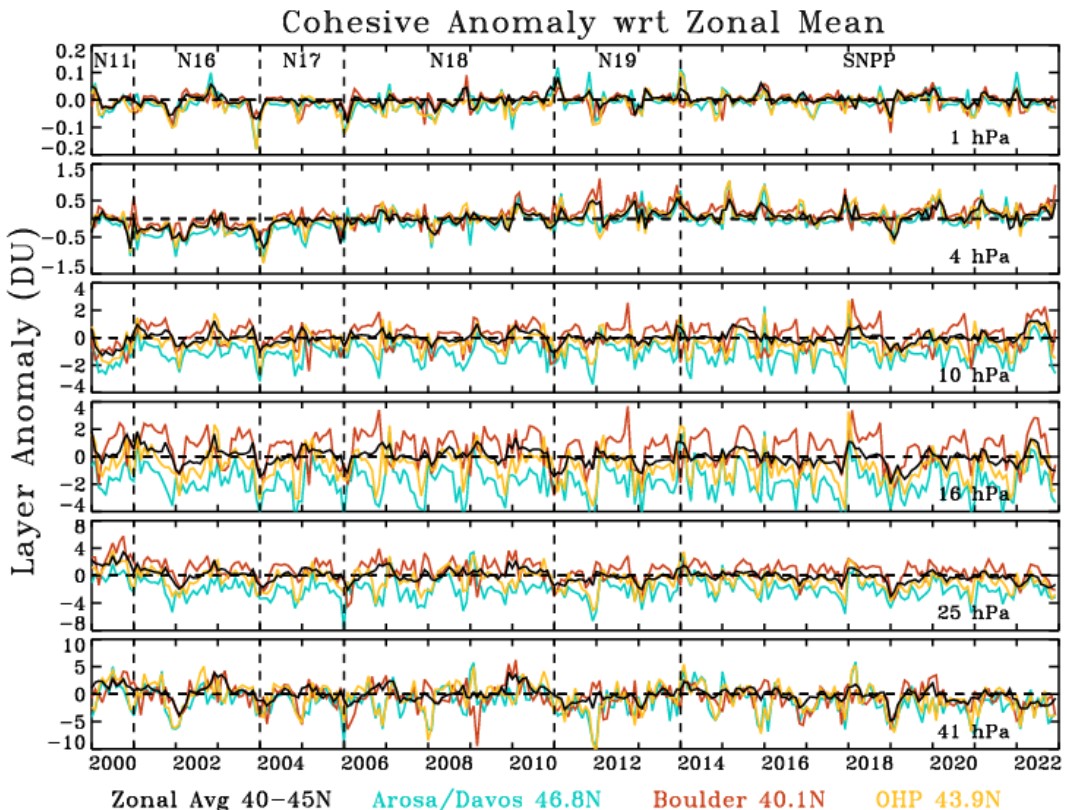

**Figure 1. Monthly ozone anomaly relative to the zonal mean monthly averages. This process leaves intact the trend for each site and the zone, and accentuates the differences between the station values since all anomalies are referenced to the zonal product. Evident at 4 hPa is a positive trend from 2002 to 2013, then a levelling out after.**

Figure 2 also shows the anomalies for the 40–45° N zonal average with the station anomalies, but each anomaly is now created using the climatology derived from each separate dataset. This removes the bias between the stations and the zones. At 1 hPa, Arosa/Davos appears to display the most variation (largest peaks and dips) in the anomalies. Since the anomaly for each site is now based on the seasonality of each site's data the structure in the anomalies is more uniform. For example, now at 16 hPa, the difference between Boulder and the two sites Arosa/Davos and OHP in the latter half of the year is removed. In 2012, where the Boulder anomaly was positive with respect to the zonal average seasonal value, and the Arosa/Davos and OHP sets were negative with respect to the zonal seasonal average, all are now of the same sign with respect to their own seasonal averages. Nonetheless, there are events where one station shows opposite anomalies to the other two, for example early 2009 at 41 hPa, when the Boulder anomaly is negative, and Arosa/Davos and OHP are positive. Thus, it is noted that when comparing daily or monthly data values from GB and satellite data, the overpass data will reveal a different structure than the zonal data. The trend calculations in this paper are based on the datasets of Fig. 2, where the seasonal behavior is removed using the station seasonality.

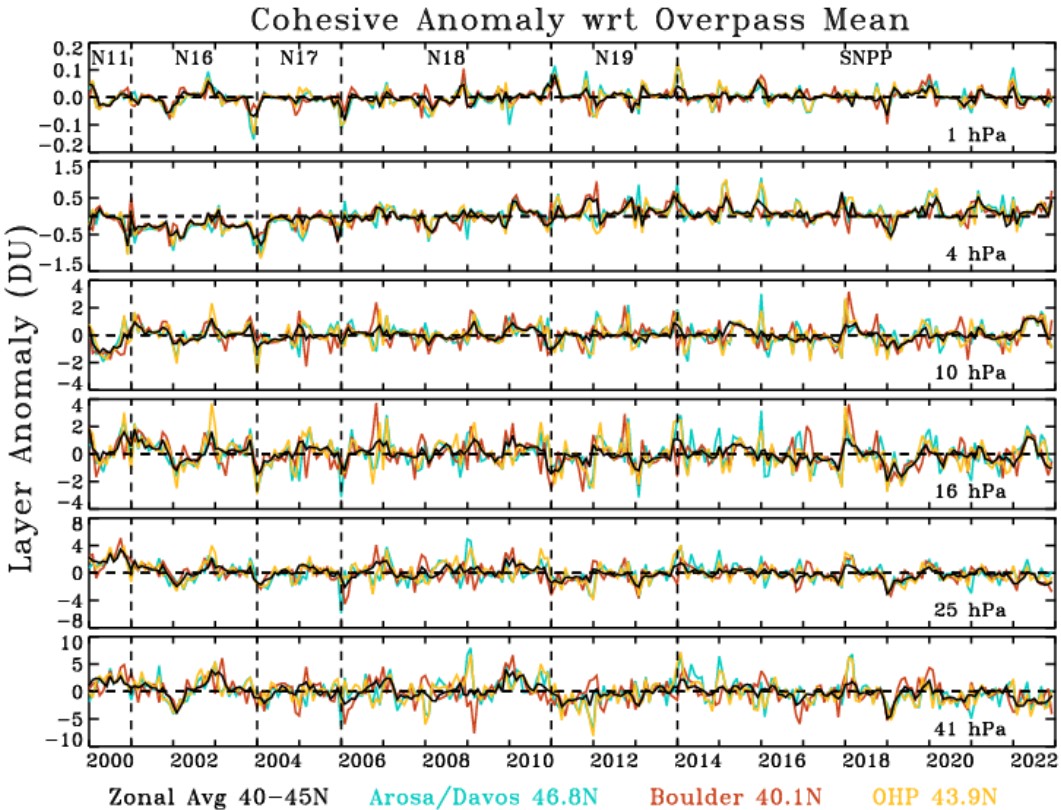

**Figure 2. Monthly ozone anomaly relative to the monthly climatology for each station overpass dataset. This process leaves intact the trend for each site and the zone, and shows the consistency among the stations when each station climatology is removed. This dataset is used for the trends calculations. Evident at 4 hPa is a positive trend from 2002 to 2013, then a leveling out after. Trends are run on this dataset.**

The COH overpass and zonal datasets have a similar vertical granularity as the Umkehr dataset, but use somewhat different pressures for the demarcation of the top and bottom of each layer. Since no additional smoothing is required, we simply use interpolation and integration to convert the COH layer profiles to the Umkehr layers. We exclude layers 1 to 4 since there is little sensitivity in SBUV and OMPS NP in these layers (Kramarova et al., 2013b). The overpass monthly-mean dataset in this study uses all COH data matched to dates when Umkehr data also exists. This dataset is publicly available at https://gml.noaa.gov/aftp/ozwv/Publications/2023_Umkehr_Ozone_Trends_Paper/. Appendix D explores the impact of temporal sampling on trends.

This study also uses a specialized zonal monthly-mean COH product which is the average of all daily profiles with an Umkehr match at the associated GB station. Zones used for most stations are the 5° wide zone which includes the geographic station latitude (Arosa/Davies: 47.5° N, OHP: 42.5° N, MLO: 17.5° N). Boulder and Lauder, however, are located directly on the border of two zones, so the zonal product in this study is the mathematical average of the two adjacent zones (Boulder: 37.5° N and 42.5° N, Lauder 42.5° S and 47.5° S).



## 3 Methods

### 3.1 LOTUS Model overview - the Reference Model

The Long-term Ozone Trends and Uncertainties in the Stratosphere (LOTUS) activity is a project of SPARC (Stratosphere-Troposphere Processes and their Role in Climate) and has produced a statistical Multiple Linear Regression (MLR) model called the LOTUS model (https://usask-arg.github.io/lotus-regression/index.html). The 2019 LOTUS report (SPARC/IO3C/GAW, 2019) and update (Godin-Beekmann et al., 2022) have quantified stratospheric ozone trends and evaluated their uncertainties. The LOTUS model is a general-least-squares approach MLR model. This study uses version 1 (v 0.8.0) with the independent linear trends (ILT) configuration. The independent linear terms represent the ozone depletion period (pre-1997), the ozone recovery period (post-2000) and an optional gap period (1997–2000). We will call the terms "pre", "post" and "gap" for short. The version 0.8.0 adds an option to enforce continuity across the gap period which is used in this study. The regression uses an interactive procedure (Cochrane and Orcutt, 1949) and the autocorrelation coefficient is adjusted with each iteration. The covariance matrix is modified accordingly to account for measurement gaps (Savin and White, 1978).

The LOTUS model (further referred as reference model in this study) is written here:

$$\hat{y}(t,z) = \beta_0(t,z)C_{pre}(t) + \beta_1(t,z)C_{post}(t) + \beta_2(t,z)Linear_{pre}(t) + \beta_3(t,z)Linear_{post}(t) + \sum_{i=4}^{n} \beta_i X_i(t,z) + \epsilon(t,z) \quad (1)$$

where $\hat{y}(t,z)$ is the estimated ozone at time $t$ and altitude $z$; β are the fitted coefficients of the model; the residual term, $\varepsilon(t, z)$ is the difference between the LOTUS model and the input data. $C_{pre}$ and $C_{post}$ are the constant terms as defined by:

$$Constant_{pre} = \begin{cases} 1 & \text{for } t < 1997\text{-Jan} \\ 1 - mt & \text{for } 1997\text{-Jan} \le t < 2000\text{-Jan} \\ 0 & \text{for } t \ge 2000\text{-Jan} \end{cases}$$

$$Constant_{post} = \begin{cases} 0 & \text{for } t < 1997\text{-Jan} \\ mt & \text{for } 1997\text{-Jan} \le t < 2000\text{-Jan} \\ 1 & \text{for } t \ge 2000\text{-Jan} \end{cases}$$

where 0.29135 and t = month starting in January 1980 and ending in December 2020. Indeed, the constant terms are only constant in the "pre" and "post" periods. The 3-year "gap" period is represented by a line of slope m connecting the two constant (pre and post period bias) terms.

The linear terms of the model are defined as:

$$linear_{pre} = \begin{cases} mt - b & \text{for } t < 1997\text{-Jan} \\ 0 & \text{for } t \ge 1997\text{-Jan} \end{cases}$$

$$linear_{post} = \begin{cases} 0 & \text{for } t < 2000\text{-Jan} \\ mt & \text{for } t \ge 2000\text{-Jan} \end{cases}$$

where m=0.008487, b = -1.700240, and t = month starting in January 1980 and ending in December 2020.

Natural variability is a complicating factor in deriving trends associated with the changes in the ozone depleting chemistry. LOTUS fits predictor variables as proxies for natural variability to the ozone data so that one can interpret the resulting linear trend as a trend due to the changes in chemistry. The summation term is the summation of the predictors used as a proxy for the dynamical induced ozone variability.



The natural variability proxies in the LOTUS model v 0.8.0 are Aerosol Optical Depth (AOD), El Nino/ Southern
Oscillation (ENSO), and the Quasi-Biennial Oscillation (QBO) in the form of the first two principal components (also
known as an empirical orthogonal function analysis). The data sources for each are described in Table 3.
Large $SO_2$ levels after volcanic eruptions can impact the validity of sonde ozone retrievals (Yoon et al., 2022). Both
Umkehr and satellite ozone profiles from SBUV and OMPS are highly uncertain and/or biased because of high aerosol
load during volcanic eruptions (DeLuisi et al, 1989; Petropavlovskikh et al., 2005, 2022; Bhartia et al, 1993, Torres
et al., 1995, Bhartia et al, 2013).  It is recommended that the data for 2 to 3 years after the El-Chichon and Pinatubo
large volcanic eruptions should not be used in trend analyses. Therefore, we exclude data during the volcanic periods
(1982–1983 and 1991–1993) from the analyzed time series. Moreover, this study is focused on the linear trend
analyses after 2000 when there are no large stratospheric aerosol perturbations that significantly influence
stratospheric ozone variability over the middle latitudes and therefore impact trend and uncertainty estimates. Since
we have eliminated the data during the volcanic period, this study does not include the AOD proxy in the calculations.
We define the 'reference' model (RM) as the proxies most commonly used for the dynamical proxies which is
equivalent to the LOTUS model v 0.8.0 minus the AOD term.  The representative equation is:
$$\sum_{i=4}^{n} \beta_i(t,z)X_i(t) = \beta_4(t,z)QBO_A(t) + \beta_5(t,z)QBO_B(t) + \beta_6(t,z)ENSO(t) + \beta_7(t,z)Solar(t)$$
(2)

The Quasi-biennial Oscillation (QBO) is derived from the Singapore radiosonde profiles (1979–2020) that detect
variability in the direction of the tropical winds in the lower stratosphere. It also shows that zonal wind variation
propagates downward with an average period of ~28 months [Wallace, 1973]. The principal component analysis of
the 100–10 hPa zonal winds can describe the majority of the wind variability. The reference model (and LOTUS v
0.8.0) use the two leading modes of the calculated empirical orthogonal functions (EOF) for trend analyses [Wallace
et al., 1993].
The El Niño/ Southern Oscillation (ENSO) is a periodic mode of climate variability of the atmosphere and sea surface
temperatures associated with the equatorial Pacific Ocean with periods ranging from 2–8 years. The Multivariate
ENSO index (MEI) is produced by the NOAA Physical Sciences Laboratory and is derived from the EOF analysis of
sea surface temperature, sea level pressure, outgoing terrestrial radiation, and surface winds in the area of the Pacific
basin from 30° S to 30° N and from 100° E to 70° W (Wolter and Timlin, 2011). Temperature anomalies in the
troposphere with corresponding stratospheric temperature anomalies during El Niño/ La Niña events modulate the
tropical upwelling of the Brewer-Dobson circulation (BDC) and thus the meridional transport of ozone in the
stratosphere. (Diallo et al., 2018).
The solar cycle is the 11-year periodic cycle of solar activity and solar irradiance that reaches the Earth's atmosphere.
The change in UV radiation that is absorbed by the atmosphere, most notably in the upper stratosphere, leads to
changes in atmospheric temperature and the photochemistry which produces ozone. (Lee and Smith, 2003). The 10.7
cm solar radio flux data is used as the proxy for the solar cycle in the LOTUS model.
Seasonal components in the form of Fourier harmonics were added into the LOTUS model with version 0.8.0. Godin-
Beekman et al. (2022) showed in their Fig. 7 that the model fit for the ozone profile satellite and model records is
improved by adding seasonal components to the proxies, increasing the adjusted R-squared ($R^2$) from 0.3 or less to
0.3 to 0.5. The seasonality and relevant contributions of some predictor's variables are compensated in this study by





adding the seasonal components to the fitted predictors. Seasonal components are represented in the model by sine

and cosine functions with periods of 12 and 6 months that describe the variability of the proxies on these timescales.

So, for each fitted predictor in the model

$$\beta_i X_i(t,z) \text{ where } i > 1$$

a seasonal variation in the form of Fourier components is added as follows:

$$\beta_m(t,z) = \beta_{m,0}(z) + \sum_{i=1}^{2} \beta_{m,1,i}(z) sin\left(\frac{2\pi it}{12}\right) + \sum_{i=1}^{2} \beta_{m,2,i}(z) cos\left(\frac{2\pi it}{12}\right)$$

### 3.2 The Extended Model - Adding Predictors

Recent publications (i.e. Petropavlovskikh et al., 2019; Szelag et al., 2020; Godin-Beekmann et al, 2022; Millan et al. 2024) highlight the need to reduce the trend uncertainties in the lower stratosphere (LS). There is still a discrepancy between modeled and observed ozone trends in the LS but large uncertainties make comparisons difficult. In this study, we test additional predictors in the model to account for dynamical variability of ozone in the stratosphere, thus improving the model performance and reducing the uncertainty of the trends. The argument for additional predictors is that the LOTUS model was developed for the regression of zonally averaged ozone data, which reduces some variability that might be impacting the ground-based records on regional bases. Impact of additional proxies in trend analyses were reported in other publications (Weber et al, 2022a, Bernet, 2023 and references therein), and were mostly found to improve the statistical model fit at high latitudes where the impact of the descending branch of the Brewer-Dobson circulation and Arctic/Antarctic oscillations has contributed to additional variability in stratospheric ozone records.

In what we define as the 'extended' model, we add single additional predictors (one at a time) in the model as such:

$$\sum_{i=4}^{n} \beta_i(t,z) X_i(t,z) = \beta_4(t,z) QBO_A(t) + \beta_5(t,z) QBO_B(t) + \beta_6(t,z) ENSO(t) + \beta_7(t,z) Solar(t) + \beta_8(t,z) X_{predictor}(t,z)$$

The fitted predictors contain Fourier components, like in the reference model, to allow for seasonal variation.

We test the following additional predictors as described below to assess the impact on trends and uncertainties:

- Quasi-Biennial Oscillation (QBO): Two notable disruptions to the otherwise relatively periodic QBO have occurred during the study period: 2015–2016 and in 2020 (Diallo, et. al 2022). Two additional leading modes of the calculated empirical orthogonal functions (EOF) are tested to improve the trend model fit during the anomalous QBO years.
- Arctic/Antarctic Oscillation (AO/AAO): the pattern of surface air pressure anomalies in the polar region and certain mid-latitude regions. The AO/AAO has strong correlations (Lawrence et al 2020) with stratospheric ozone through the strength of the polar vortex. The positive phase of the AO or AAO in the winter months is associated with low activity in the vertically propagating planetary Rossby waves, a strong polar vortex, a low vortex wavenumber, and low stratospheric temperatures. Thus, the positive (negative) phase of the AO/AAO is correlated to low (high) ozone anomalies especially in the winter months (Lawrence et al, 2020).



● North Atlantic Oscillation (NAO): Similar to the Arctic Oscillation, this is a pattern of surface air pressure
anomalies between certain regions in the high altitudes of the North Atlantic Ocean. This index is calculated
by the pressure difference between the Azores high and the subpolar low.
● Eddy Heat Flux (EHF): The flux of heat through a zonal plane by transport due to the Brewer-Dobson
circulation, here averaged from 45–75° N/S (use EHF S for Lauder only). This represents the planetary wave
activity that drives transport of ozone.
● Tropopause Pressure (TP): Pressure level of boundary between the troposphere and the stratosphere. In this
study, we use the monthly mean pressure level of the tropopause from the NOAA National Centers for
Environmental Prediction (NCEP) reanalysis product. As the troposphere warms due to release of GHGs
and the stratosphere cools due to ODSs destroying stratospheric ozone, the tropopause is rising (Meng et al.,
2021). Thompson et al, (2021) and Stauffer et al., (2023) found that the lower stratospheric ozone trends in
tropics become slightly positive when recomputed with respect to the tropopause height (which has its own
trend). This finding indicates that ozone depletion in the lower stratosphere (i.e. Ball et al., 2020) is driven
by climate-change-related changes in transport and mixing in the lower stratosphere. Therefore, we are
testing the TP proxy in the model to account for non-chemical ozone losses in order to assess chemical
attribution of ozone trends.
● Equivalent Latitude (EqLat): Geographical latitude of the isoline encircling the area of equal Potential
Vorticity (PV) (Lary et al, 1995). The EqLat normalizes the range of PV values that change with season and
interannual and makes it convenient for interpretation of ozone variability and trends (i.e. Wohltmann et al
2005). The dataset was generated from GMI CTM analyses (private communications with Susan Strahan,
June 2021) for each ground-based station overpass criteria (latitude and longitude envelope, see above) and
at several altitude levels coincident with Umkehr ozone profile layers. Appendix C discusses a COH dataset
based on EqLat instead of geometric latitude. No advantage was found by using the EqLat coordinate system
for the COH zonal dataset.

Source datasets for all predictors in the reference and extended models are shown in Table 3.

| Predictor | Description | Source |
|---|---|---|
| **ENSO** | **El Nino/Southern Oscillation** | **Monthly Mean Multivariate ENSO Index https://psl.noaa.gov/enso/mei.old/**[1] |
| **Solar** | **Solar 10.7cm flux** | **https://spaceweather.gc.ca/forecast-prevision/solar-solaire/solarflux/sx-5-en.php** |





| QBO | Quasi-Biennial Oscillation | **Principal Component Analysis of the Monthly Mean Zonal Wind https://www.geo.fu-berlin.de/met/ag/strat/produkte/qbo/qbo.dat** |
|---|---|---|
| **AOD** | **AOD is included in the LOTUS model, but not used in this study** | |
| AO | Arctic Oscillation, Monthly Mean index | http://www.cpc.ncep.noaa.gov/products/precip/CWlink/daily_ao_index/monthly.ao.index.b50.current.ascii |
| AAO | Antarctic Oscillation, Monthly Mean index | https://www.cpc.ncep.noaa.gov/products/precip/CWlink/daily_ao_index/aao/aao.shtml |
| NAO | The North Atlantic Oscillation, monthly mean index | https://www.cpc.ncep.noaa.gov/products/precip/CWlink/pna/norm.nao.monthly.b5001.current.ascii.table |
| EHF | Eddy Heat Flux | Cumulative Mean (from September to April) of Heat Flux at 100 hPa from MERRA2 reanalysis averaged over 45–75° N (45–75° S for Lauder), deseasonalized. It is kept constant from April to Sep. https://acd-ext.gsfc.nasa.gov/Data_services/met/ann_data.html |
| TP | Tropopause Pressure | Monthly Mean NCEP-NCAR reanalysis (Kalnay et al., 1996); Tropopause pressure at the lat/lon of each station, deseasonalized. ftp://ftp.cdc.noaa.gov/Datasets/ncep.reanalysis.derived/tropopause/pres.tropp.mon.mean.nc |
| EqLat | Equivalent Latitude | Monthly Mean equivalent latitude derived from MERRA2 -GMI CTM potential vorticity (PV) contours on 31 potential temperature surfaces [Susan Strahan, private communication, 8/24/2022]. The PV at each station is determined by a 1/distance weighted average of the values in a ± 2 lat, ± 2 lon grid, then converted to EqLat on the Umkehr layers. |



**Table 3: List of predictors either previously used (bolded) in the LOTUS 0.8.0 (reference) model and additional predictors**
**evaluated in this study for a future use in the extended LOTUS trend regression model. Note, two components of the QBO**
**predictors were used in the reference model (i.e. Godin-Beekmann et al., 2022). We added two more components in the**
**extended model for tests described in this paper.**
[1] **Since the incorporation of the ENSO index into the LOTUS model, NOAA GSL has updated the index to v1.2.**
**https://psl.noaa.gov/enso/mei/. However, for consistency with results from the Godin-Beekmann (2022) paper we use the**
**old MEI index that is part of the LOTUS v 0.8.0 package.**
All proxies are used as is. No de-trending (removal of the long-term trend in proxy) is applied to the proxies. Therefore,
we interpret any changes to the trends derived with additional proxies as approximations of trends driven by chemistry
and transport related to climate change. These are rough approximations as some feedbacks are known to impact
chemistry (e.g. changes in stratospheric temperature).

## 380 3.3 The Full Model - Combining Additional Predictors

After we have determined the impact of the additional predictors singly, we discern which predictors should be
combined to constitute the 'Full Model'. Prior to selecting additional predictors for the 'Full Model', we perform
correlation tests to identify any cross correlations between predictors. We select predictors that are not highly
correlated (less than +/- 0.2) to ensure that all predictors are largely independent. We use the square of the Pearson
correlation coefficient $R2$ for each pair of the predictors to test our assumptions. We find that ENSO, Solar, QBO
(1,2,3,4), AAO, AO, EHF (N and S), and TP (at each station) have correlations less than +/- 0.2 (with the exception
of $R2 = 0.3$ for EHF (N) and AO). Therefore, any of these predictors can be combined in the 'Full Model'. We find
that NAO has a correlation of .38 with AO so we do not use these two predictors in the same model.
We also test the independence of EqLat proxies calculated at several geographic locations (defined by the latitude and
longitude of each Umkehr station) and by selecting a proxy at several altitude levels centered in the middle of Umkehr
layers 3–9. We find that the $R2$ between the TP and EqLat in the lower stratosphere (Umkehr layer 3) can be large but
anticorrelated -0.7 (Boulder), moderate 0.4 (MLO and Lauder), while close to zero at Arosa/Davos and low at OHP
(-0.2). In the middle and upper stratosphere, the $R2$ varies from -0.5 to -0.4 (MLO), 0.2 to 0.3 (Arosa/Davos and
OHP), 0.5 to 0.6 (Boulder), and 0.4 to 0.7 (Lauder). EqLat has mostly low correlations ($< \pm 0.3$) with all other proxies
except for higher correlations with QBO B in layers 5 (-0.3) and 6 (-0.4), and QBO A in layer 7 (0.3) at MLO; and
with AO in layer 8 (0.3) at OHP and Arosa/Davos. Also, EqLat has no correlation with the TP proxy in layer 4 in
Boulder, in layer 9 at Lauder, and in layers 8 and 9 at OHP. Since there are occasional high correlations between
EqLat and TP proxies, we do not use them together in the 'Full trend Model'.

## 399 4 Results

### 400 4.1 Reference Model Trend Results

First, we discuss the reference model trends derived from the COH overpass, Umkehr and ozonesonde records at 5
geographic locations. All datasets are deseasonalized with a climatology computed from a subset of data taken from
1998–2008 prior to the trend analysis. Trend results are presented in Fig. 3 and organized in 5 panels. Each panel
shows trends at selected pressure/altitude levels detected from Umkehr (green), COH (orange) and ozonesonde (blue)



records at Arosa/Davos, OHP, Boulder, MLO/Hilo and Lauder ground-based stations. Ozonesonde data for the
Arosa/Davos panel are selected from Hohenpeißenberg, Germany station that is in close vicinity to Arosa/Davos
station. We show trends for layers where the measurement is of highest quality: Umkehr (layer 3 through 8), COH
(layers 5 through 9) and ozonesonde (layers 3 through 5) records.
The Umkehr data used in this analysis is the monthly mean of all available Umkehr data (one or two measurements
per day). The sonde and COH monthly means use only those profiles that have corresponding Umkehr measurements
on that date. We explore the impact of temporal sampling on trends in Appendix D. For COH with the Umkehr
matched data, trends are slightly larger at OHP but well within the error bars. At all other stations the COH trends are
not impacted by sampling. At OHP the ozonesonde trends matched to Umkehr are slightly larger at layer 4 only and
well within the error bars; while at Lauder in layers 4 and 5 trends are smaller, but barely within the error bars.
In the upper (above 10 hPa) stratosphere, Umkehr (green) and COH (orange) trends are positive and agree within the
error bars (+/- 2 standard errors). The exception is found at 8–2 hPa pressure level over the Lauder station, where
Umkehr trends are near zero and COH trends are ~ +3–4 %/decade. The error bars show +/- 2 standard errors, and the
fact that they do not overlap suggests that the differences in trends are statistically significant. This could be related
to the relatively large uncertainties in the instrumental corrections applied to homogenize the Umkehr record
(Petropavlovskikh et al, 2022). Björklund (2023) discusses relative drifts in Umkehr, ozonesonde, FTIR and MW
ozone records over Lauder. The authors are not able to identify instrumental artifacts that may have caused the
discrepancies in the co-located records, but point out that it is not related to the sampling biases.
In the middle stratosphere (60–10 hPa) agreement between Umkehr and COH is within uncertainty of the trend except
at Arosa/Davos where COH trends are statistically different from Umkehr trends at 16–8 hPa. COH trends at 32–16
hPa are mostly negative (-2–3 %/decade) with the exception of Lauder where trends are near zero and similar to
Umkehr trends. Umkehr trends between 32–16 hPa are close to zero. The ozonesonde trends (blue) agree with COH
(orange) and Umkehr (green) trends in layer 63–16 hPa at Arosa/Davos, Boulder and MLO. However, at OHP
(Lauder) the ozonesonde trends are found to be positive at +3±3 %/decade (negative at -3±1.5 %/decade) and
significantly different from the near-zero trends seen in the COH and Umkehr results.
In the lower stratosphere (125–63 hPa), Umkehr trends vary between small positive (+1–2 %/decade at Hilo and
Lauder) and negative (-2–3 %/decade at Arosa/Davos, OHP and Boulder); however, trend uncertainties are the largest
(2 standard errors are 2–3 %/decade, (see Table 4 below) in comparison to the middle and upper stratospheric trends.
Ozonesonde trends at OHP station are positive (+4 %/decade), and negative over Lauder (-2 %/decade). They also
feature large uncertainties (±5 %/decade) that are larger than the uncertainties found in Umkehr trends which could
be caused by the limited sampling (see Appendix D). Sonde trends at Hilo show negative trend values with large
uncertainties. But the data in this study at Hilo is not corrected for the ozonesonde drop off after 2014 known to occur
at this station (Stauffer, 2022), so the deviation from the Umkehr results at these levels may be misleading.
Figure 3 also shows trends derived from the zonal-mean COH data associated with each station (orange dashed line).
These are shown for comparison with the overpass COH data (solid line) to study the impact of the spatial sampling
biases on the trends. Though Figs. 1 and 2 show clear interannual differences between the records from the individual
stations, and the associated zonal average, we find very small differences in trends (mostly in the upper stratosphere





at middle latitude stations). Therefore, the station overpass sampling provides trends that are representative of the
zonal averaged trends (Zerefos, 2018) and the discrepancies in trends between GB and satellite records do not strongly
depend on the spatial sampling differences.

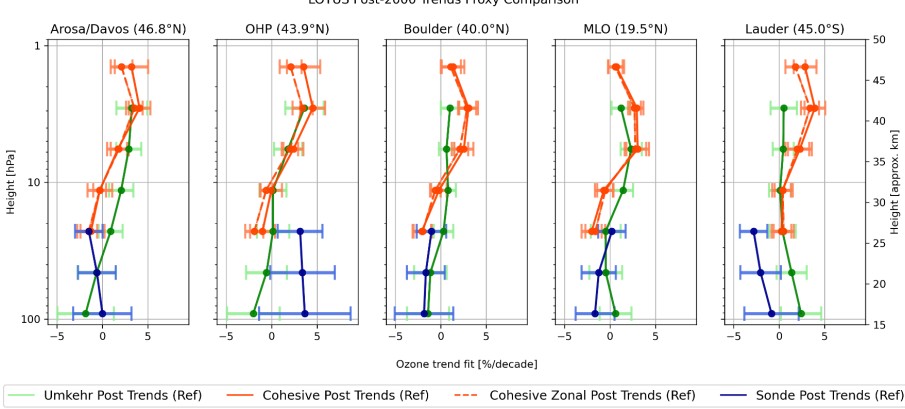


**Figure 3: The 2000–2020 ozone trends are shown at 7 altitude/pressure levels. The LOTUS model v 0.8.0 is used for trend**
**analyses. Umkehr trends (green), COH (orange) and ozonesonde (blue) are shown for 5 ground-based stations:**
**Arosa/Davos, OHP, Boulder, MLO and Lauder (panels left to right). Ozonesonde data for the Arosa/Davos panel are**
**selected from Hohenpeißenberg, Germany that is in close vicinity to Arosa/Davos. Trends from the zonal-mean COH data**
**(orange dashed line) are shown for comparison with the overpass COH data (solid line). The error bars indicate ± 2 standard**
**errors.**
**4.2 Standard Error of Reference Model**

| LOTUS Model Proxy Tests: Standard Error for Reference Model | | | | | | | | | | | | | | | |
|---|---|---|---|---|---|---|---|---|---|---|---|---|---|---|---|
| **Height** | **Umkehr** | **Arosa/Davos** | | | **OHP** | | | **Boulder** | | | **MLO** | | | **Lauder** | | |
| **(hPa)** | **Layer** | **UMK** | **COH** | **SND** | **UMK** | **COH** | **SND** | **UMK** | **COH** | **SND** | **UMK** | **COH** | **SND** | **UMK** | **COH** | **SND** |
| **1-2** | **9** | | 0.92 | | | 0.91 | | | 0.62 | | | 0.43 | | | 0.63 | |
| **2-4** | **8** | 0.85 | 0.59 | | 1.06 | 0.68 | | 0.51 | 0.52 | | 0.52 | 0.37 | | 0.72 | 0.57 | |
| **4-8** | **7** | 0.69 | 0.59 | | 0.77 | 0.54 | | 0.41 | 0.52 | | 0.58 | 0.62 | | 0.57 | 0.66 | |
| **8-16** | **6** | 0.66 | 0.68 | | 0.75 | 0.59 | | 0.42 | 0.43 | | 0.55 | 0.49 | | 0.61 | 0.56 | |
| **16-32** | **5** | 0.66 | 0.75 | 0.76 | 0.89 | 0.68 | 1.26 | 0.54 | 0.51 | 0.77 | 0.82 | 0.55 | 0.75 | 0.73 | 0.54 | 0.73 |
| **32-63** | **4** | 1.05 | | 1.04 | 1.13 | | 1.95 | 0.90 | | 1.04 | 0.90 | | 0.94 | 0.83 | | 1.16 |
| **63-127** | **3** | 1.55 | | 1.60 | 0.15 | | 2.75 | 1.15 | | 1.63 | 0.87 | | 1.07 | 1.11 | | 1.50 |


**Table 4: Standard Error (SE) for the Reference model 2000–2020 trend for five ground-based station locations**
**(Arosa/Davos, OHP, Boulder, MLO and Lauder). Results are provided for trend analyses of the Cohesive satellite (COH),**
**Dobson Umkehr (UMK) and ozonesonde (SND) records and for Umkehr. The layers are selected to represent the best**
**quality of data. Values of SE shown are the actual errors in DU.**
We will use the standard error of the trend fit to the data to evaluate the improvements in the model fit after additional
proxies are included. We use the standard error as a metric instead of standard deviation to reduce dependence on the





number of points in the trends model. The **Table 4** provides the Standard Errors for the Reference Model fit and
represents uncertainty of the trend in DU of the mean ozone in each layer at the station. The standard errors of the
trend detected in three co-located ozone records at each station (or in the nearby location as in case of Arosa/Davos
or MLO comparisons) do not significantly differ, although in general ozonesonde errors are slightly larger than
Umkehr errors most likely due to the larger sampling errors in ozonesonde monthly mean record. Also, the errors in
trends detected in COH layers 5–8 are on average smaller than for Umkehr trends (with the exception of layer 7 at
Boulder, MLO and Lauder) which could be explained by an overpass method that averages several satellite profiles
from adjacent orbits and therefore reduces meteorological scale variability in averaged ozone data.

### 4.3 Adjusted R2

The adjusted R2 values of the 2000–2020 trends are shown in Fig. 4 and Table 5 for the data fit using the Reference
model. The adjusted R2 is a modified version of R2 that adjusts for the number of predictors in a regression model
and represents the 'goodness' of the model fit to the data.  For COH adjusted R2 is shown for both the overpass and
the zonal datasets.
Though values are significantly less than the high values usually seen when comparing data that includes the prevalent
seasonal variation, the adjusted R2 values for the COH zonal mean record are similar in magnitude and vertical shape
to the results of the (60°S–60°N) broadband trend analyses published in Godin-Beekmann (2022), Fig. 7 varying
between 0.1 and 0.5.  We designate the average values (0.3) as a threshold for satisfactory fit indicating conformance
with prior LOTUS results.  We indicate in bold in Table 5 adjusted R2 values of 0.3 or greater to note achievement of
that threshold.
The adjusted R2 for the Reference model fit is slightly better for the zonal mean COH data than for the COH overpass
over the Northern middle latitude stations. This is expected as much of the variability of the time series is reduced in
the zonal average as compared to the station overpass data as shown in Fig. 2, and more easily explained by the
typically used predictors. Indeed, the goal of this study is to determine if the additional predictors help to explain the
additional variation as measured at point locations.
The model fit to the GB data is similar to the COH overpass results in the middle stratosphere (layers 5 and 6), but the
model explains less ozone variability in the Umkehr records in the upper stratosphere (layers 7 and 8). In the lower
stratosphere (layers 3, 4 and 5), the model fit to the ozonesonde and Umkehr records is similar with the exception of
Lauder (Umkehr has larger adjusted R2 in layers 4 and 5).  The adjusted R2 for COH overpass in layer 5 is similar to
Umkehr and sonde with a larger difference at OHP. The adjusted R2 in the lower stratosphere is less than in the middle
stratosphere, which points to other processes (e.g., transport) that drive ozone variability. In this paper we investigate
improvement to the trend model fit by introducing additional proxies that can improve representation of the
dynamically-driven ozone variability in the stratosphere.



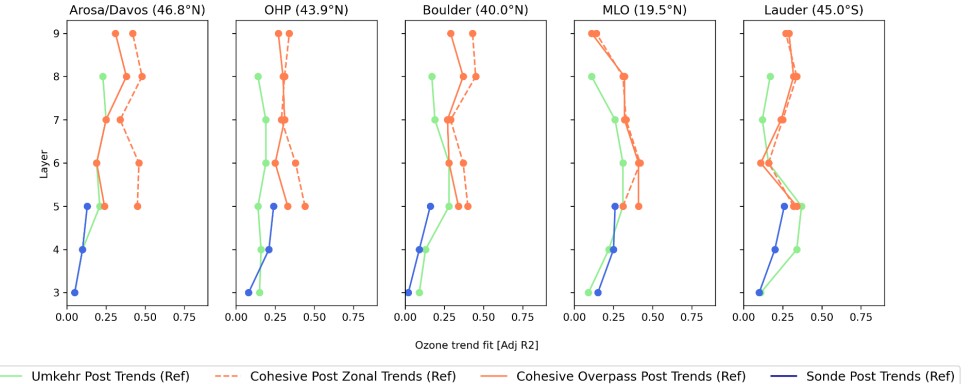


**Figure 4: The adjusted R2 is plotted as a function of altitude/pressure for the LOTUS model fit to the Umkehr (green), ozonesonde (blue), COH overpass (orange, solid), and COH zonal-mean (orange, dashed). Results are shown in 5 panels that represent trend analyses of ozone records over Arosa/Davos (Hohenpeißenberg for sondes), OHP, Boulder, MLO (Hilo for sondes) and Lauder ground-based stations.**

| LOTUS Model Proxy Tests: Adjusted R2 for Reference Model | | | | | | | | | | | | | | | | |
|---|---|---|---|---|---|---|---|---|---|---|---|---|---|---|---|---|
| **Height** | **Umkehr** | **Arosa/Davos** | | | **OHP** | | | **Boulder** | | | **MLO** | | | **Lauder** | | |
| **(hPa)** | **Layer** | **UMK** | **COH** | **SND** | **UMK** | **COH** | **SND** | **UMK** | **COH** | **SND** | **UMK** | **COH** | **SND** | **UMK** | **COH** | **SND** |
| **1-2** | **9** | | **0.31** | | | 0.27 | | | 0.29 | | | 0.11 | | | 0.29 | |
| **2-4** | **8** | 0.23 | **0.38** | | 0.14 | **0.30** | | 0.17 | **0.37** | | 0.11 | **0.32** | | 0.17 | **0.32** | |
| **4-8** | **7** | 0.25 | 0.25 | | 0.19 | **0.31** | | 0.19 | 0.27 | | 0.26 | **0.32** | | 0.12 | 0.24 | |
| **8-16** | **6** | 0.19 | 0.19 | | 0.19 | 0.25 | | 0.28 | 0.28 | | **0.31** | **0.41** | | 0.16 | 0.11 | |
| **16-32** | **5** | 0.21 | 0.24 | 0.13 | 0.14 | **0.33** | 0.24 | 0.28 | **0.34** | 0.16 | **0.31** | **0.41** | 0.25 | **0.37** | **0.34** | 0.26 |
| **32-63** | **4** | 0.10 | | 0.10 | 0.16 | | 0.21 | 0.13 | | 0.09 | 0.22 | | 0.24 | **0.34** | | 0.20 |
| **63-127** | **3** | 0.05 | | 0.05 | 0.15 | | 0.08 | 0.09 | | 0.02 | 0.09 | | 0.14 | 0.11 | | 0.10 |

**Table 5: Similar to Table 4, but for the adjusted R2. Values of 0.30 and above are indicated in Bold as a threshold to**
**indicate a satisfactory fit.**
**4.4 Reference Model P-Values:**
The p-values are often used to evaluate statistical significance of predicted results and results labelled "significant" if
they remain below a threshold of 0.05. However, Chang et al. (2021) argued as Wasserstein et al. (2019) does that all
trends should be reported with their associated p-values and a thorough discussion of the certainty of trend detection
as described by the p-values. Therefore, the p-values can be used for understanding the certainty of the trend. Under
the IGAC TOAR activity, p-values are scored to define a consistent scale for comparison of the trends between
different analyses (see Table 3, Chang et al., 2023).

| LOTUS Model Proxy Tests: Reference Model P Value | | | | | | | | | | | | | | | | |
|---|---|---|---|---|---|---|---|---|---|---|---|---|---|---|---|---|
| **Height** | **Umkehr** | **Arosa/Davos** | | | **OHP** | | | **Boulder** | | | **MLO** | | | **Lauder** | | |
| **(hPa)** | **Layer** | **UMK** | **COH** | **SND** | **UMK** | **COH** | **SND** | **UMK** | **COH** | **SND** | **UMK** | **COH** | **SND** | **UMK** | **COH** | **SND** |





| | | | | | | | | | | | | | | | | |
|---|---|---|---|---|---|---|---|---|---|---|---|---|---|---|---|---|
| 1-2 | 9 | | 0.00 | | | 0.00 | | | 0.03 | | | 0.10 | | | 0.00 | |
| 2-4 | 8 | 0.00 | 0.00 | | 0.00 | 0.00 | | 0.05 | 0.00 | | 0.02 | 0.00 | | 0.47 | 0.00 | |
| 4-8 | 7 | 0.00 | 0.00 | | 0.02 | 0.00 | | 0.12 | 0.00 | | 0.00 | 0.00 | | 0.43 | 0.00 | |
| 8-16 | 6 | 0.00 | 0.62 | | 0.84 | 0.98 | | 0.05 | 0.66 | | 0.01 | 0.17 | | 0.85 | 0.50 | |
| 16-32 | 5 | 0.17 | 0.08 | 0.05 | 0.87 | 0.15 | 0.01 | 0.58 | 0.00 | 0.04 | 0.56 | 0.00 | 0.93 | 0.61 | 0.62 | 0.00 |
| 32-63 | 4 | 0.55 | | | 0.57 | 0.62 | | 0.06 | 0.21 | | 0.11 | 0.61 | | 0.11 | 0.10 | 0.03 |
| 63-127 | 3 | 0.23 | | 1.00 | 0.17 | | 0.15 | 0.22 | | 0.25 | 0.47 | | 0.08 | 0.03 | | 0.49 |

**Table 6: Similar to Table 4 but for p-values. Values of less than 0.05 (high certainty of trend detection) are shown in green. Values between .05 and 0.1 (yellow) medium certainty, between 0.1 and 0.33 (orange) low certainty of trend detection, and above 0.33 (red) very low certainty or no evidence of trend detection.**

Table 6 provides p-values for the Reference Model. These are further used as a baseline for comparison to model fits with additional predictors. P-values of the reference model fit suggest a high certainty (p<0.05) in detected trends in the COH data in layers 7, 8 and 9 at almost all stations with the exception of the higher p-value (0.1, medium certainty) found at MLO in layer 9. Also, high certainty in derived trends is reached for COH records in layer 5 at Boulder and MLO.

Umkehr trend analyses also show high confidence in trend detection at Arosa/Davos and MLO stations in layers 6, 7 and 8, at OHP in layers 6 and 8, and in Boulder in layers 7 and 8. For the ozonesonde data the high confidence (i.e. low uncertainty) is found for Hohenpeißenberg, OHP, and Boulder trends detected in layer 5, and at Lauder in layers 4 and 5.

The medium level of the certainty (0.05<p≤0.10) is found in trends detected in layer 5 of COH ozone time series at Arosa/Davos, layer 4 of ozonesonde records at OHP, layer 9 COH and layer 3 of ozonesonde at MLO, and in layer 4 of Umkehr at Lauder.

Low certainty in detected trends at p-value of 0.10 (not inclusive) to 0.33 is found in Umkehr layer 3 and 5 at Arosa/Davos; in COH layer 5, Umkehr and ozonesonde layer 3 at OHP; in Umkehr layers 3, 4 and 7, and ozonesonde layers 3 and 4 at Boulder; and in ozonesonde layer 4 and COH layer 6 record at MLO.

Highest (lowest certainty) p-values (>0.33) were found in layer 6 of COH overpass records at most stations (except for MLO where p-values are medium high). We note that the COH trends are close to zero and the uncertainty envelope crosses the zero line. Therefore, these results point to the trend model's inability to detect non-zero trends and account for all ozone variability in this layer. Similarly, near-zero Umkehr trends with relatively large SE in layer 6 at OHP and Lauder, layer 5 at all (except Arosa/Davos) stations, and in layer 3 and 4 at MLO show the same level of high p-values thus suggesting that additional proxies should be added in the trend model to assess the impacts of the natural variability and instrumental noise on trend uncertainty.

It is also important to note that the reference trend model fit to ozone in Umkehr layers 7 and 8 at Lauder has high p-values, which is related to the near-zero trends that shows large disagreement with COH trend. This difference could be caused by remaining instrumental step changes that were not fully removed during the record homogenization (Petropavlovskikh et al., 2022).





While near-zero trends and high p-values are found in the fit of the Hilo ozonesonde record in layer 5, the p-values in
layers 3 and 4 show only medium p-values for near zero trends. It is possible that infrequent launches of ozonesonde
observations at Hilo could create the temporal sampling bias and appear noisy. The ozonesonde record at
Hohenpeißenberg has sufficiently frequent sampling (3 times per week) for successful trend analyses (Chang et al.,
2020; Chang, 2023 preprint), but the p-values remain high in layers 3–4. The p-values for Umkehr fit at Arosa/Davos
are in the medium to high range for layers 3,4,5, but somewhat smaller which could be due to non-zero trends in layers
3 and 5. The p-value difference could be also related to the different location of the ozonesonde (HOH) and Umkehr
(Arosa/Davos) observations, thus the records could contain different atmospheric variability that might impact the
model fit.
We will discuss changes to the p-values in the next section after we add more proxies to the trend model in an attempt
to improve confidence in trend detection.
**5 Trends with the Extended Model - testing the addition of single predictors**
The LOTUS styled Reference Model is developed and optimized for zonal average datasets. Modeling and trend
analysis for GB and satellite overpass data may improve by the addition of other proxies not used in the reference
model to improve capturing processes that impact ozone changes over limited geographical regions. The Extended
Model tests the addition of single predictors to see if fit statistics can be improved for GB and overpass datasets. We
judge success of the Extended Model by examination of the reduction in the Standard Error of the trend term, and by
evaluation of the impact on the adjusted R2 of the model fit. Table 7 displays the change in the Standard Error of the
post 2000 trend for each proxy tested determined as $SE_{ref}$ - $SE_{ext}$ as a percent of $SE_{ref}$. As such positive values
correspond to the desired reduction of SE, and are highlighted in the table in green. Low impact changes in the SE
are highlighted in yellow, and increases in SE (negative values) are highlighted in orange, or red. It may seem unusual
for the addition of proxies to increase the SE (negative values in the table) which indicates less confidence in the fit.
But these SE are the uncertainty in the trend term, not in the overall model fit. The new proxies considered each have
a possible trend and associated error budget for that trend. Whether the additional proxy increases trend uncertainty
can depend on how well the trend of the new proxy can be characterized. The adjusted R2 is a better indicator of the
overall model improvement. Table 8 displays the adjusted R2 for the Extended Model for each proxy tested. Values
of 0.30 and above are indicated in bold as a threshold to indicate a satisfactory fit.

| a)    LOTUS Model Proxy Tests: Adding Tropopause Pressure (% difference in Std. Error of Model) | | | | | | | | | | | | | | | | |
|---|---|---|---|---|---|---|---|---|---|---|---|---|---|---|---|---|
| Height | Umkehr | Arosa/Davos | | | OHP | | | Boulder | | | MLO | | | Lauder | | |
| (hPa) | Layer | UMK | COH | SND | UMK | COH | SND | UMK | COH | SND | UMK | COH | SND | UMK | COH | SND |
| 1-2 | 9 | | 0.33 | | | 0.11 | | | 0.49 | | | 1.39 | | | 3.01 | |
| 2-4 | 8 | -0.71 | -0.51 | | -0.09 | -0.44 | | -0.19 | 0.38 | | -0.58 | -0.27 | | 1.26 | 2.62 | |
| 4-8 | 7 | -0.29 | 0.00 | | 0.26 | 1.30 | | 0.25 | -0.19 | | 2.61 | 0.33 | | 3.71 | 1.36 | |
| 8-16 | 6 | -1.07 | -0.73 | | 0.00 | 0.34 | | 0.72 | -0.23 | | 0.55 | 0.82 | | 3.11 | 5.39 | |
| 16-32 | 5 | -0.15 | 2.14 | -0.93 | 1.13 | 5.34 | 2.28 | -0.37 | 0.59 | 0.60 | 4.54 | 9.31 | 2.72 | 0.00 | 0.74 | 2.44 |



| 32-63 | 4 | 6.60 | | 6.07 | 5.87 | | 9.81 | 3.35 | | 7.46 | 7.02 | | 6.05 | 8.03 | | 9.44 |
| 63-127 | 3 | 12.80 | | 10.17 | 12.80 | | 10.91 | 6.81 | | 6.01 | 5.77 | | 4.55 | 9.76 | | 7.94 |


| b) LOTUS Model Proxy Tests: Adding Equivalent Latitude (% difference in Std. Error of Model) | | | | | | | | | | | | | | | | |
|---|---|---|---|---|---|---|---|---|---|---|---|---|---|---|---|---|
| Height | Umkehr | Arosa/Davos | | | OHP | | | Boulder | | | MLO | | | Lauder | | |
| (hPa) | Layer | UMK | COH | SND | UMK | COH | SND | UMK | COH | SND | UMK | COH | SND | UMK | COH | SND |
| 1-2 | 9 | | 8.37 | | | 2.85 | | | 1.94 | | | -7.18 | | | 2.85 | |
| 2-4 | 8 | -0.47 | 0.68 | | 0.09 | 1.17 | | -0.39 | 1.53 | | -3.48 | -5.42 | | 0.98 | 3.14 | |
| 4-8 | 7 | 3.75 | 3.19 | | 2.08 | 0.56 | | 5.41 | 4.08 | | -2.61 | -3.90 | | 0.53 | 1.21 | |
| 8-16 | 6 | 6.11 | 8.28 | | 2.54 | 10.88 | | 2.39 | 7.75 | | 5.29 | 7.76 | | 3.44 | 7.72 | |
| 16-32 | 5 | 7.93 | 10.55 | 5.87 | 1.92 | 13.35 | 7.97 | -1.85 | 0.00 | 2.84 | 0.25 | 0.73 | 0.65 | 0.82 | 3.91 | -0.77 |
| 32-63 | 4 | -1.44 | | -1.80 | 3.20 | | 0.39 | -0.22 | | -0.10 | 0.33 | | 1.03 | -0.24 | | 0.44 |
| 63-127 | 3 | 1.29 | | 2.02 | -1.43 | | -3.26 | -0.79 | | -0.79 | 9.57 | | 2.32 | 1.36 | | 1.88 |


| c) LOTUS Model Proxy Tests: Adding QBO CD (% difference in Std. Error of Model) | | | | | | | | | | | | | | | | |
|---|---|---|---|---|---|---|---|---|---|---|---|---|---|---|---|---|
| Height | Umkehr | Arosa/Davos | | | OHP | | | Boulder | | | MLO | | | Lauder | | |
| (hPa) | Layer | UMK | COH | SND | UMK | COH | SND | UMK | COH | SND | UMK | COH | SND | UMK | COH | SND |
| 1-2 | 9 | | -3.70 | | | 1.54 | | | -1.30 | | | -3.70 | | | -4.91 | |
| 2-4 | 8 | 0.24 | 5.09 | | 5.65 | 13.78 | | 0.00 | 5.74 | | -4.26 | -1.90 | | 0.00 | -2.27 | |
| 4-8 | 7 | -2.74 | -1.34 | | 2.20 | 0.37 | | -2.21 | -2.33 | | -2.26 | -3.25 | | -0.18 | -0.60 | |
| 8-16 | 6 | -3.51 | -4.51 | | -3.07 | -3.57 | | -0.72 | 0.94 | | -0.91 | 0.20 | | -2.13 | -3.95 | |
| 16-32 | 5 | -2.59 | -1.34 | -0.40 | -3.84 | -1.78 | 1.14 | -1.48 | -0.59 | -1.94 | 14.23 | 9.67 | 12.55 | -1.09 | 0.74 | 1.67 |
| 32-63 | 4 | -1.44 | | 0.28 | -2.22 | | -1.18 | 0.00 | | 0.10 | 9.81 | | 7.38 | -0.84 | | -2.56 |
| 63-127 | 3 | -3.04 | | -1.78 | -1.09 | | -1.00 | -0.17 | | -0.79 | -0.58 | | 2.41 | 0.00 | | -3.63 |


| d) LOTUS Model Proxy Tests: Adding AO/AAO (% difference in Std. Error of Model) | | | | | | | | | | | | | | | | |
|---|---|---|---|---|---|---|---|---|---|---|---|---|---|---|---|---|
| Height | Umkehr | Arosa/Davos | | | OHP | | | Boulder | | | MLO | | | Lauder | | |
| (hPa) | Layer | UMK | COH | SND | UMK | COH | SND | UMK | COH | SND | UMK | COH | SND | UMK | COH | SND |
| 1-2 | 9 | | 1.20 | | | -1.64 | | | 0.32 | | | -1.85 | | | -0.48 | |
| 2-4 | 8 | -0.83 | 0.00 | | -3.77 | -1.17 | | -0.78 | -0.38 | | -2.13 | -2.44 | | 0.84 | -1.92 | |
| 4-8 | 7 | -0.72 | 1.68 | | -4.15 | -2.60 | | 3.19 | 4.66 | | 1.22 | -3.41 | | 1.24 | -1.21 | |
| 8-16 | 6 | -0.15 | -0.58 | | -2.41 | -3.91 | | 1.20 | 0.47 | | 1.64 | -1.63 | | -0.33 | 2.51 | |
| 16-32 | 5 | -1.22 | -0.40 | -1.20 | 0.45 | -2.08 | -2.28 | 0.19 | -0.59 | -2.09 | 3.93 | 1.82 | 0.65 | -1.64 | -1.49 | 1.41 |
| 32-63 | 4 | 5.84 | | 7.78 | 0.36 | | 5.47 | -1.23 | | -1.71 | 7.58 | | 1.74 | -0.72 | | 2.47 |
| 63-127 | 3 | 13.12 | | 12.86 | 5.45 | | 8.00 | -1.40 | | -3.22 | 4.38 | | 1.34 | -1.08 | | 2.42 |


| e) LOTUS Model Proxy Tests: Adding NAO (% difference in Std. Error of Model) | | | | | | | | | | | | | | | | |
|---|---|---|---|---|---|---|---|---|---|---|---|---|---|---|---|---|
| Height | Umkehr | Arosa/Davos | | | OHP | | | Boulder | | | MLO | | | Lauder | | |
| (hPa) | Layer | UMK | COH | SND | UMK | COH | SND | UMK | COH | SND | UMK | COH | SND | UMK | COH | SND |





| Height (hPa) | Umkehr Layer | Arosa/Davos UMK | COH | SND | OHP UMK | COH | SND | Boulder UMK | COH | SND | MLO UMK | COH | SND | Lauder UMK | COH | SND |
|---|---|---|---|---|---|---|---|---|---|---|---|---|---|---|---|---|
| 1-2 | 9 | | 0.54 | | | -2.52 | | | -0.16 | | | -3.70 | | | -1.27 | |
| 2-4 | 8 | -0.24 | 0.00 | | -3.11 | -1.91 | | -0.39 | -1.91 | | -1.74 | -3.79 | | -1.96 | -1.75 | |
| 4-8 | 7 | -0.58 | 0.67 | | -1.95 | -2.04 | | 0.00 | 3.88 | | 2.61 | -1.14 | | -2.83 | -2.41 | |
| 8-16 | 6 | 0.15 | -0.87 | | -1.74 | -3.40 | | -2.15 | -2.82 | | 2.37 | -0.41 | | -1.97 | -0.54 | |
| 16-32 | 5 | -0.46 | -1.20 | -1.07 | 0.68 | -2.23 | -3.98 | -0.37 | -1.38 | -1.20 | -1.35 | -0.73 | -2.98 | -2.46 | -4.28 | -1.54 |
| 32-63 | 4 | 2.58 | | 3.13 | -0.62 | | -0.39 | -0.22 | | 0.40 | 1.45 | | -0.82 | -2.64 | | -4.77 |
| 63-127 | 3 | 10.60 | | 6.74 | 2.65 | | 1.67 | 0.44 | | -2.73 | 1.73 | | -0.45 | -2.26 | | -4.91 |


| f) LOTUS Model Proxy Tests: Adding Eddy Heat Flux (% difference in Std. Error of Model) | | | | | | | | | | | | | | | | |
|---|---|---|---|---|---|---|---|---|---|---|---|---|---|---|---|---|
| Height (hPa) | Umkehr Layer | Arosa/Davos UMK | COH | SND | OHP UMK | COH | SND | Boulder UMK | COH | SND | MLO UMK | COH | SND | Lauder UMK | COH | SND |
| 1-2 | 9 | | 4.89 | | | 4.61 | | | 4.38 | | | -3.24 | | | 0.16 | |
| 2-4 | 8 | -1.42 | 4.58 | | 2.64 | 6.01 | | 3.12 | 8.80 | | -1.55 | -2.98 | | 0.70 | 1.92 | |
| 4-8 | 7 | -2.74 | -3.36 | | -0.39 | -3.90 | | -2.95 | -2.33 | | 5.04 | -4.39 | | 1.77 | 4.52 | |
| 8-16 | 6 | -3.21 | -3.20 | | -2.54 | -4.76 | | -2.39 | -3.52 | | -1.09 | 0.41 | | -0.16 | 1.08 | |
| 16-32 | 5 | -3.35 | -2.80 | -3.20 | -2.15 | -3.71 | -2.28 | -2.59 | -2.37 | -2.09 | 9.33 | -0.36 | 4.14 | 0.68 | 2.42 | 0.90 |
| 32-63 | 4 | -1.91 | | -1.61 | -2.04 | | -1.97 | -2.79 | | -2.82 | 8.70 | | 2.97 | 1.92 | | 2.21 |
| 63-127 | 3 | 1.49 | | 1.35 | -0.88 | | -1.79 | -2.53 | | -2.61 | 0.92 | | 0.80 | 2.08 | | 1.68 |

**Table 7: Change in Standard Error (SE) of the post-2000 trend estimate, in percent of SE of Reference Model for adding**
**single predictors. Panel a: Tropopause Pressure; b: Equivalent Latitude; c: QBO terms C and D; d: AO/AAO; e: NAO; f:**
**Eddy Heat Flux.**

| a) LOTUS Model Proxy Tests: Adding Tropopause Pressure (Adjusted R2 of Model) | | | | | | | | | | | | | | | | |
|---|---|---|---|---|---|---|---|---|---|---|---|---|---|---|---|---|
| Height (hPa) | Umkehr Layer | Arosa/Davos UMK | COH | SND | OHP UMK | COH | SND | Boulder UMK | COH | SND | MLO UMK | COH | SND | Lauder UMK | COH | SND |
| 1-2 | 9 | | **0.31** | | | 0.27 | | | 0.29 | | | 0.11 | | | **0.31** | |
| 2-4 | 8 | 0.23 | **0.38** | | 0.15 | **0.30** | | 0.17 | **0.38** | | 0.10 | **0.31** | | 0.18 | **0.34** | |
| 4-8 | 7 | 0.24 | 0.24 | | 0.19 | **0.32** | | 0.19 | 0.27 | | 0.28 | **0.32** | | 0.14 | 0.24 | |
| 8-16 | 6 | 0.19 | 0.19 | | 0.19 | 0.25 | | 0.29 | 0.28 | | **0.32** | **0.42** | | 0.19 | 0.15 | |
| 16-32 | 5 | 0.21 | 0.26 | 0.13 | 0.15 | **0.39** | 0.26 | 0.27 | **0.35** | 0.16 | **0.34** | **0.47** | 0.28 | **0.36** | 0.35 | 0.30 |
| 32-63 | 4 | 0.21 | | 0.22 | 0.29 | | **0.32** | 0.19 | | 0.18 | 0.29 | | 0.29 | **0.42** | | **0.31** |
| 63-127 | 3 | 0.24 | | 0.23 | **0.42** | | 0.21 | 0.22 | | 0.11 | 0.14 | | 0.18 | 0.25 | | 0.21 |


| b) LOTUS Model Proxy Tests: Adding Equivalent Latitude (Adjusted R2 of Model) | | | | | | | | | | | | | | | | |
|---|---|---|---|---|---|---|---|---|---|---|---|---|---|---|---|---|
| Height (hPa) | Umkehr Layer | Arosa/Davos UMK | COH | SND | OHP UMK | COH | SND | Boulder UMK | COH | SND | MLO UMK | COH | SND | Lauder UMK | COH | SND |
| 1-2 | 9 | | **0.43** | | | **0.37** | | | **0.36** | | | 0.15 | | | **0.32** | |
| 2-4 | 8 | 0.23 | **0.39** | | 0.14 | **0.31** | | 0.17 | **0.39** | | 0.10 | **0.30** | | 0.18 | **0.34** | |
| 4-8 | 7 | **0.35** | **0.34** | | **0.31** | **0.41** | | 0.27 | **0.33** | | 0.29 | **0.36** | | 0.17 | 0.27 | |
| 8-16 | 6 | **0.31** | **0.35** | | **0.33** | **0.45** | | **0.33** | **0.40** | | **0.40** | **0.51** | | 0.25 | 0.23 | |
| 16-32 | 5 | **0.34** | **0.39** | 0.26 | 0.25 | **0.51** | 0.33 | **0.31** | **0.40** | 0.18 | **0.31** | **0.42** | 0.26 | **0.42** | **0.41** | 0.29 |





| Height (hPa) | Umkehr Layer | Arosa/Davos UMK | COH | SND | OHP UMK | COH | SND | Boulder UMK | COH | SND | MLO UMK | COH | SND | Lauder UMK | COH | SND |
|---|---|---|---|---|---|---|---|---|---|---|---|---|---|---|---|---|
| 32-63 | 4 | 0.11 | | 0.09 | 0.19 | | 0.21 | 0.12 | | 0.08 | 0.22 | | 0.25 | **0.34** | | 0.21 |
| 63-127 | 3 | 0.08 | | 0.07 | 0.16 | | 0.08 | 0.12 | | 0.02 | 0.18 | | 0.19 | 0.14 | | 0.12 |


| c) LOTUS Model Proxy Tests: Adding QBO CD (Adjusted R2 of Model) | | | | | | | | | | | | | | | | |
|---|---|---|---|---|---|---|---|---|---|---|---|---|---|---|---|---|
| Height | Umkehr | Arosa/Davos | | | OHP | | | Boulder | | | MLO | | | Lauder | | |
| (hPa) | Layer | UMK | COH | SND | UMK | COH | SND | UMK | COH | SND | UMK | COH | SND | UMK | COH | SND |
| 1-2 | 9 | | **0.31** | | | **0.30** | | | **0.31** | | | 0.10 | | | 0.28 | |
| 2-4 | 8 | 0.25 | **0.44** | | 0.18 | **0.43** | | 0.19 | **0.44** | | 0.09 | **0.32** | | 0.19 | **0.33** | |
| 4-8 | 7 | 0.24 | 0.26 | | 0.22 | **0.34** | | 0.20 | 0.28 | | 0.25 | **0.31** | | 0.13 | 0.25 | |
| 8-16 | 6 | 0.19 | 0.18 | | 0.18 | 0.24 | | **0.30** | **0.32** | | **0.32** | **0.43** | | 0.17 | 0.10 | |
| 16-32 | 5 | 0.22 | 0.24 | 0.15 | 0.13 | **0.34** | 0.27 | 0.29 | **0.36** | 0.16 | **0.40** | **0.48** | **0.35** | **0.38** | **0.37** | **0.31** |
| 32-63 | 4 | 0.12 | | 0.13 | 0.17 | | 0.22 | 0.14 | | 0.10 | 0.29 | | **0.32** | **0.35** | | 0.20 |
| 63-127 | 3 | 0.05 | | 0.07 | 0.17 | | 0.09 | 0.11 | | 0.02 | 0.10 | | 0.18 | 0.13 | | 0.10 |


| d) LOTUS Model Proxy Tests: Adding AO/AAO (Adjusted R2 of Model) | | | | | | | | | | | | | | | | |
|---|---|---|---|---|---|---|---|---|---|---|---|---|---|---|---|---|
| Height | Umkehr | Arosa/Davos | | | OHP | | | Boulder | | | MLO | | | Lauder | | |
| (hPa) | Layer | UMK | COH | SND | UMK | COH | SND | UMK | COH | SND | UMK | COH | SND | UMK | COH | SND |
| 1-2 | 9 | | **0.33** | | | 0.26 | | | **0.32** | | | 0.11 | | | **0.30** | |
| 2-4 | 8 | 0.23 | **0.39** | | 0.13 | **0.30** | | 0.18 | **0.38** | | 0.11 | **0.31** | | 0.18 | **0.31** | |
| 4-8 | 7 | 0.24 | 0.26 | | 0.20 | **0.31** | | 0.23 | **0.33** | | 0.29 | **0.32** | | 0.14 | 0.23 | |
| 8-16 | 6 | 0.20 | 0.19 | | 0.18 | 0.24 | | **0.31** | 0.30 | | **0.34** | **0.42** | | 0.19 | 0.16 | |
| 16-32 | 5 | 0.22 | 0.24 | 0.13 | 0.15 | **0.33** | 0.24 | 0.29 | **0.34** | 0.15 | 0.34 | **0.44** | 0.27 | **0.37** | **0.34** | 0.28 |
| 32-63 | 4 | 0.17 | | 0.19 | 0.18 | | 0.28 | 0.13 | | 0.09 | **0.30** | | 0.28 | **0.33** | | 0.22 |
| 63-127 | 3 | 0.18 | | 0.19 | 0.24 | | 0.18 | 0.09 | | 0.02 | 0.15 | | 0.18 | 0.11 | | 0.14 |


| e) LOTUS Model Proxy Tests: Adding NAO (Adjusted R2 of Model) | | | | | | | | | | | | | | | | |
|---|---|---|---|---|---|---|---|---|---|---|---|---|---|---|---|---|
| Height | Umkehr | Arosa/Davos | | | OHP | | | Boulder | | | MLO | | | Lauder | | |
| (hPa) | Layer | UMK | COH | SND | UMK | COH | SND | UMK | COH | SND | UMK | COH | SND | UMK | COH | SND |
| 1-2 | 9 | | **0.33** | | | 0.26 | | | **0.31** | | | 0.11 | | | **0.30** | |
| 2-4 | 8 | 0.24 | **0.39** | | 0.13 | **0.30** | | 0.19 | **0.37** | | 0.12 | **0.31** | | 0.17 | **0.32** | |
| 4-8 | 7 | 0.25 | 0.26 | | 0.18 | **0.31** | | 0.20 | **0.32** | | **0.30** | **0.34** | | 0.11 | 0.23 | |
| 8-16 | 6 | 0.21 | 0.20 | | 0.18 | 0.24 | | 0.28 | 0.27 | | **0.35** | **0.44** | | 0.16 | 0.12 | |
| 16-32 | 5 | 0.22 | 0.23 | 0.14 | 0.16 | **0.33** | 0.23 | 0.28 | **0.34** | 0.17 | **0.31** | **0.42** | 0.24 | **0.36** | **0.33** | 0.28 |
| 32-63 | 4 | 0.14 | | 0.14 | 0.17 | | 0.24 | 0.15 | | 0.13 | 0.25 | | 0.25 | **0.33** | | 0.19 |
| 63-127 | 3 | 0.15 | | 0.13 | 0.20 | | 0.14 | 0.12 | | 0.03 | 0.12 | | 0.16 | 0.10 | | 0.09 |


| f) LOTUS Model Proxy Tests: Adding Eddy Heat Flux (Adjusted R2 of Model) | | | | | | | | | | | | | | | | |
|---|---|---|---|---|---|---|---|---|---|---|---|---|---|---|---|---|
| Height | Umkehr | Arosa/Davos | | | OHP | | | Boulder | | | MLO | | | Lauder | | |
| (hPa) | Layer | UMK | COH | SND | UMK | COH | SND | UMK | COH | SND | UMK | COH | SND | UMK | COH | SND |



| 1-2 | 9 | | **0.39** | | | **0.34** | | | **0.35** | | | 0.11 | | | 0.29 | |
| 2-4 | 8 | 0.25 | **0.44** | | 0.17 | **0.38** | | 0.23 | **0.46** | | 0.12 | **0.31** | | 0.18 | **0.33** | |
| 4-8 | 7 | 0.24 | 0.24 | | 0.21 | **0.31** | | 0.20 | 0.28 | | **0.30** | **0.31** | | 0.13 | 0.28 | |
| 8-16 | 6 | 0.19 | 0.19 | | 0.18 | 0.24 | | 0.28 | 0.28 | | **0.31** | **0.42** | | 0.16 | 0.13 | |
| 16-32 | 5 | 0.21 | 0.24 | 0.14 | 0.15 | **0.34** | 0.24 | 0.28 | **0.34** | 0.17 | **0.36** | **0.42** | 0.29 | **0.37** | **0.35** | 0.27 |
| 32-63 | 4 | 0.11 | | 0.11 | 0.17 | | 0.20 | 0.13 | | 0.09 | 0.28 | | 0.28 | **0.35** | | 0.21 |
| 63-127 | 3 | 0.07 | | 0.09 | 0.18 | | 0.07 | 0.09 | | 0.01 | 0.11 | | 0.17 | 0.12 | | 0.11 |

**Table 8: Adjusted R2 after adding single predictors. Panel a: Tropopause Pressure; b: Equivalent Latitude; c: QBO terms C and D; d: AO/AAO; e: NAO; f: Eddy Heat Flux.**

**5.1 Tropopause pressure (TP)**

Adding the TP proxy to the standard LOTUS model produces the most consistent results between different techniques (COH, Umkehr and ozonesonde) and also have similar magnitude of standard error changes among different latitudes (i.e. Arosa/Davos, OHP, Boulder, MLO, Lauder). The most significant impact in improving the SE is found in the lower stratosphere (layers 3, 4) and in the middle stratosphere (layer 5) at the MLO tropical station. The impact of the TP proxy on the COH trend uncertainty in the model stratosphere (layer 5) is somewhat larger, likely due to the satellite AK extending into the lower stratosphere. Similarly, larger reduction of the standard error in the Umkehr trends in the lowermost stratosphere (layer 3) in comparison to the AK-smoothed ozonesonde record could be due to sampling biases in the ozonesonde record. Adding the TP proxy to the Reference Model improves the adjusted R2 in layers 3–5, whereas the SE improvements are also consistent across geo-locations and measurement techniques. Several improvements resulted in adjusted R2 to exceed the 0.3 threshold (Umkehr at OHP in layer 3, sonde at OHP in layer 4, and sonde at Lauder in layers 3 and 4) and in many cases the adjusted R2 increased by more than 0.02.

**5.2 Equivalent Latitude (EqLat)**

In the mid-latitudes, the addition of EqLat as a predictor shows consistent results across measurement techniques and stations with few exceptions. The reduction in the SE of the model fit is evident in the COH data in the upper stratosphere (above 4 hPa or ~ 40 km), but is less pronounced in Umkehr profiles. The impact on MLO SE of the trend fit in the upper stratosphere is negative (in both COH and Umkehr records) which can be explained by the fact that the EqLat is much closer to geometric latitude near the equator than at the middle/high latitudes and therefore its use as a proxy would not provide any additional information in interpretation of the tropical upper stratospheric ozone variability. It could also suggest that the addition of EqLat will overfit the record.

The ozone record trend fits in the middle stratosphere (32–4 hPa or 25–40 km) benefit from adding the EqLat proxy at most locations. Improvement in the SE of the trends in the lower stratosphere (127–63 hPa or ~15–20 km) is minimal, limited to some locations and instrumental records (Arosa/Davos Umkehr and HOH sonde, MLO Umkehr and sonde, and Lauder Umkehr and ozonesonde), which could be related to the location of subtropical jet that modulates mixing of tropical and subtropical (and occasionally polar) air masses and influences the strat/trop exchange. Unexpectedly, the addition of the EqLat proxy to the MLR statistical model for trend detection in Boulder Umkehr and ozonesonde low stratospheric ozone records increases the uncertainties of the fit, while the influence of



subtropical jet on Boulder lower stratosphere is well known (Manney et al, 2018). Perhaps, the data analyses also
need to consider the tropopause variability.
In terms of the impact on the adjusted R2, the EqLat proxy significantly improves model fit for multiple instruments,
mostly in layers 5–7, and in COH fit in layer 9. The adjusted R2 improvements also often exceeded 0.3 threshold. No
significant improvement is found in the ozonesonde model fit in layer 5 with the exception of the OHP record (0.09
increase).

**5.3 Extra QBO terms C and D**

QBO is an important driver of ozone variability at tropical stations. Based on the results of adding 2 extra terms of the
QBO to the standard model, the recommendation could be to exercise this option only for the tropical station trends.
At the Northern middle latitudes (i.e in Arosa/Davos, OHP and, to a lesser degree, in Boulder) an improvement to the
trend SE uncertainties in layer 8 is noted. There seems to be a similar pattern for the upper stratosphere in trends
derived with Heat Flux. Tweedy et al. (2017) show that the first two EOFs of the QBO did not describe the anomalous
QBO behavior, while Anstey et al. (2021) show that the addition of two more EOFs of the QBO could capture the
effect of the disruptions on the zonal winds. Therefore, including additional QBO EOFS could benefit attribution of
ozone variability in the middle stratosphere (layers 4 and 5) in the tropical latitudes (reduced errors in MLO/Hilo
trends) and in the upper stratosphere (layer 8 in Umkehr and COH trends) in the NH middle latitude stations
(Arosa/Davos, OHP, Boulder) related to the global circulation pattern that are also represented by the Heat Flux proxy.
A slight reduction in the errors at SH middle latitude (COH and sonde at Lauder, New Zealand) could be invoked by
the EqLat variability that has a small correlation with the QBO-D proxy. Reduction of SE in the trend fit of the layer
5 ozonesonde record at OHP (up to 2 %) is not found in the Umkehr or COH results, which suggests overfitting and
sampling bias (see results in Appendix D).
The addition of extra QBO terms improves the adjusted R2 model fit for all COH station overpass records in layer 8
and occasionally improves Umkehr adjusted R2 (except at MLO). The most significant improvement is found at MLO
in layers 3–5 in all three instrument records.

**5.4 Arctic and Antarctic Oscillations (AO/AAO)**

AO/AAO proxies reduce SE (green colored cells) in the lower stratosphere (layers 3 and 4) at Arosa/Davos, OHP, and
MLO, although the reduction somewhat differs between the Umkehr and ozonesonde records. At the same time, at
Boulder and Lauder the SE does not show an improvement after the addition of the AO/AAO proxy (AAO is used
instead of AO at Lauder). In the middle stratosphere (layer 7), a reduction in SE is found over Boulder in both COH
and Umkehr records. The addition of AO/AAO proxies improves the SE of the trend at MLO and Lauder but only in
Umkehr records, while it worsens the COH SE. At Lauder, the COH SE in layer 6 shows an improvement, but not in
Umkehr record. Since results in the middle stratosphere (layers 5–7) are not always consistent among different
techniques (reductions are not in the same layers) it could indicate statistical model overfit into the record's noise, or
vertical smoothing of the Umkehr or COH technique that combines ozone variability in the layer with a portion of
ozone variability in the adjacent layers, thus partially or completely reducing the correlation with the proxy.



The addition of the AO predictor increases the adjusted R2 in the lower stratosphere at Arosa/Davos, OHP and MLO.
Also, a small enhancement of the adjusted R2 is seen in the middle and upper stratosphere, including in Umkehr layers
6 and 7 and COH layers 6, 7 and 9 over Boulder, as well as in Umkehr fit in layers 5–7 at MLO, and at Lauder (AAO)
for Umkehr and COH records in layer 6. These results are not very consistent across different geolocations, but seem
to be consistent across instrumental records at some stations (Umkehr and ozonesonde in the lower layers, and COH
and Umkehr in the upper layers).

### 5.5 North Atlantic Oscillation (NAO)

Including the NAO proxy in the trend model appears to have a similar pattern (i.e., in latitude and altitude) of changes
in the standard error as compared to the result of inclusion of the AO/AAO proxy. It is not a surprise, since indices of
the NAO and AO are highly correlated in time due to their common link to the downward propagation of stratospheric
anomalies. Standard errors are somewhat reduced in the lower stratospheric layers at the middle NH latitude and
tropical Umkehr records, but the change is less significant than in AO/AAO cases. The impacts on ozonesonde trend
uncertainties are very minimal and inconclusive at Boulder (layers 3 and 5), OHP (layer 4) and MLO  (layer 3) records.
The impacts on Lauder are similar or stronger (SE is increased for both Umkehr and sonde records) to the impacts of
the AO/AAO. In the middle and upper stratosphere the standard errors are typically reduced. The exception is found
in layer 7 of the COH record at Boulder and Arosa/Davos, and in layers 6 and 7 of the Umkehr record at MLO. Similar
negative results are found when AO/AAO proxies are added, which suggests that the observed time series are
overfitted and potentially some instrumental or sampling anomalies are misinterpreted with addition of these proxies.

### 5.6 Eddy Heat Flux (EHF)

The EHF represents a dynamical proxy for assessment of the impact of the Brewer Dobson Circulation (BDC). It is
expected to have an impact on the upper stratospheric ozone by accelerating the transport in the upper branch that
brings more ozone at higher latitudes (i.e. Arosa/Davos) and middle latitudes (i.e. OHP, Boulder, and Lauder). It could
possibly represent changes in the lower branch of the BDC circulation and the expansion of the tropical band, thus
modulating ozone in the lower stratosphere at tropics (i.e. MLO). In the Southern middle latitudes (i.e. Lauder), the
correlations could be related to the shift in the subtropical wave activities to the higher latitudes in response to the
ozone hole healing.
The addition of the EHF predictor leads to the reduced SE uncertainties in the upper stratosphere in COH and Umkehr
trends at OHP and Boulder, and in COH only trends at Arosa/Davos. It has a much smaller reduction of SE for the
Lauder trend and even an increase in uncertainties if used to fit upper stratospheric ozone time series at MLO. At the
same time, the SE in the Umkehr and ozonesonde middle stratosphere (layers 4–5) is substantially reduced, including
smaller improvements at Lauder. In the lower stratospheric (layer 3) ozone trend SE in Umkehr and sonde records at
MLO, Lauder and Arosa/Davos are somewhat reduced when using the EHF proxy.
Addition of the EHF predictor seems to have an impact in the upper stratosphere increasing the adjusted R2 for COH
records in layers 8 and 9 in all but MLO records, which indicates impact of the BDC upper branch on the middle
latitudes. In contrast to the COH, the Umkehr adjusted R2 has not changed significantly, which possibly suggests a



high measurement noise in the station records. There is, however, a small increase in adjusted R2 in the Umkehr
record in layer 7 at MLO (whereas COH does not show a change).
The increase in adjusted R2 is found at MLO in Umkehr and sonde layers 4 and 5, including a small increase in layer
3, which probably is related to the EHF-driven changes in the middle stratosphere . Ozone variability in Umkehr and
sonde records at MLO appears to contain information about the circulation changes in the shallow BDC branch.
**6 The Full Model - adding multiple predictors**
In this paper we seek to develop an improved model and thus trend estimates for point located measurements of ozone
through modifications of a model optimized for zonal data. Our criteria for model improvement are based on reduction
of the SE of the trend with either improvement (at best) or moderate impact (at worst) on the model adjusted R2. From
the results of the previous section, we see several opportunities to improve the model and improve confidence in the
trend estimates. This section examines if the gains of the above are improved while adding several predictors together.
As stated above the TP as a predictor exhibits the most consistent results for all stations and measurement techniques.
The other predictors have successes in SE reduction, but only at some layers, and some stations. Some results are
instrument dependent.
Based on the tests above we expect combining predictors can improve the model fit and trend SE reduction, but it is
clear that the predictor selection should vary by station and level. Appendix E details the choices made for the Full
Model which combines 1 to 3 additional proxies beyond the Reference Model.
**6.1 Predictors added for the Full Model**
Reduction of the SE of the trend while improving (or at least not impacting) model adjusted R2 is the basis of predictor
choice for the Full Model. To qualify a predictor should exhibit consistent results for all measurement techniques.
Improvement at multiple stations is preferred to single station improvements. In general, we avoid combining highly
correlated predictors. Table 9 shows final choices for the Full Model.

| LOTUS Full Model predictor selection | | | | | |
|---|---|---|---|---|---|
| | **Arosa/Davos** | **OHP** | **Boulder** | **MLO** | **Lauder** |
| **Layer** | | | | | |
| **9** | EqLat | EqLat | EqLat | Reference only | EqLat |
| **8** | EqLat | EqLat | EqLat | Reference only | EqLat |
| **7** | EqLat | EqLat | EqLat | Reference only | EqLat |
| **6** | EqLat | EqLat | EqLat | EqLat | EqLat |
| **5** | EqLat | EqLat | EqLat | EqLat, QBO CD, AO | EqLat |
| **4** | TP, AO | TP, AO | TP | TP, QBO CD, AO | TP |
| **3** | TP, AO | TP, AO | TP | TP, QBO CD, AO | TP |




**Table 9:** **Added predictors for the Full model are tuned for each layer and each station. For layers 7 to 9 the SE and adjusted R2 parameters at MLO are not improved by additional predictors, and the original LOTUS based Reference Model is used. Appendix E explains the logic of the predictor selection.**

**6.2 Impact of the Full Model on trends**

Figure 5a shows the trends for the stations (with COH overpass) for the Reference and Full Models. An impact of the Full Model on ozone trends derived in the upper stratosphere (above 16 hPa) is neutral. Addition of proxies to the LOTUS model does not change trends which remain the same magnitude as those derived using the Reference Model, i.e. positive and statistically significant at the SH and MH middle latitudes and over tropics. The largest difference (outside of the SE uncertainty) between upper stratospheric Umkehr and COH trends is found over Boulder, MLO and Lauder.

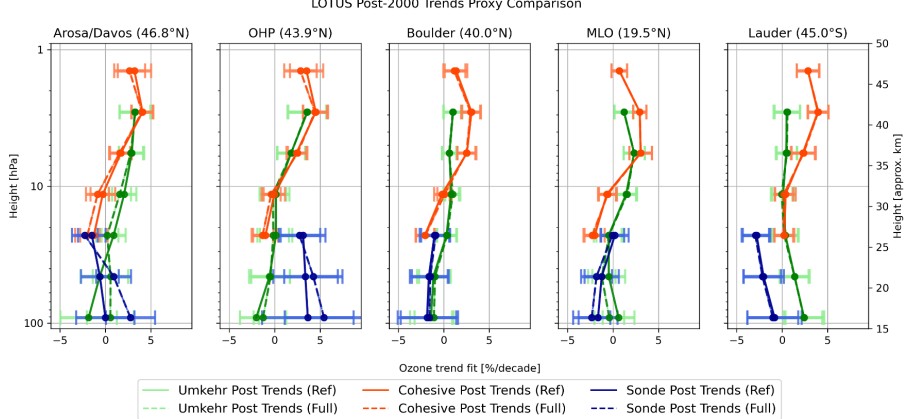

**Figure 5a: Post 2000 trends for the Full and Reference Model. In this figure the COH data shown in orange is the overpass data. Solid lines depict Reference Model values (unchanged from Fig. 3). Dashed lines depict Full Model values for all 3 instrument types.**

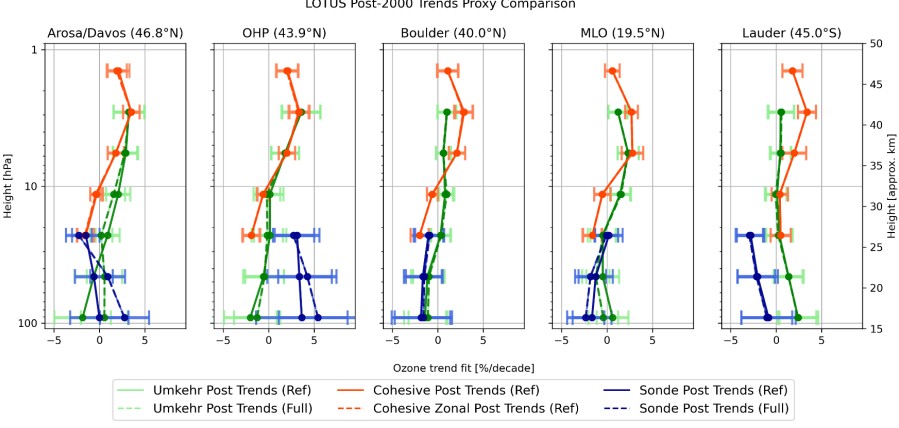



**Figure 5b: Post 2000 trends for the Full and Reference Model. In this case the orange lines are with the zonal data instead of the COH overpass data. Dashed lines depict Full Model values for all 3 instrument types. The Umkehr and sonde trends are unchanged from Fig. 5a.**

In the middle stratosphere, additional proxies do not change trend values across locations and instrumental records (outside of the SE). At OHP, Boulder and Lauder Umkehr trends in layer 6 (8–16 hPa) are barely positive while COH trends are negative. At Arosa/Davos and MLO, COH trends in layer 6 are barely negative and Umkehr trends are significantly positive. Most COH trends in layer 5 (16–32 hPa) are statistically negative (except at Lauder), while Umkehr trends are near zero.

In the lower stratosphere, Umkehr and sonde trends Arosa/Davos and MLO change after the Full model is used. However, Umkehr and sonde trend changes at MLO are within the SE and therefore can be deemed not significant. Ozonesonde trends at Arosa/Davos in layer 3 (125–63 hPa) change from zero to positive. Umkehr trends at Arosa/Davos in layer 3 change from negative to near zero. Large differences between ozonesonde and Umkehr trends at Lauder and OHP remain unchanged after the Full model is applied although respective SE envelopes overlap.

Figure 5b also shows the trends for the Reference and Full Models, but the COH data shown is the associated zonal data relevant to each station. Incorporation of the additional proxies does not change the trend values for the zonal COH data. Impact on error estimates for the trends are discussed next.

**6.2 Impact of the Full Model on the Trend SE**

Table 10 summarizes the reduction in the SE for the Full model. Selection of the EqLat predictor for the Full model in the layers 5–9 and for all stations (except MLO/Hilo, to be discussed later) shows the improvement in the SE (as discussed in the previous section). Also, the TP predictor is selected for inclusion to the Full model for trend analyses at Boulder and Lauder stations in layers 3 and 4. The combination of several predictors are used for individual stations based on the additional reduction in the SE. For the Arosa/Davos and OHP stations we select a combination of the TP and AO to reduce the SE almost twice as much in some layers. Inclusion of AO proxy is in support of the interpretation of seasonal and interannual ozone variability recorded over stations in Europe that are north of 40 degrees latitudes and are exposed to the seasonal events of ozone depleted air masses transported from the Polar region during the spring season (Steinbrecht et al., 2011; Manney et al., 2011; Knudsen and Grooss, 2000; Fioletov and Shepherd, 2003; Zhang et al., 2017; Weber et al., 2022a). The strong impact of AO/AAO on the lower stratosphere ozone variability are not detected in Boulder or Lauder and we choose not to include it in the Full model for trend analyses at these stations.

| LOTUS Model Proxy Tests: (% Difference in Std. Error of Model) | | | | | | | | | | | | | | | | |
|---|---|---|---|---|---|---|---|---|---|---|---|---|---|---|---|
| Height | Umkehr | Arosa/Davos | | | OHP | | | Boulder | | | MLO | | | Lauder | | |
| (hPa) | Layer | UMK | COH | SND | UMK | COH | SND | UMK | COH | SND | UMK | COH | SND | UMK | COH | SND |
| 1-2 | 9 | | 8.35 | | | 2.74 | | | 1.94 | | | 0.00 | | | 2.85 | |
| 2-4 | 8 | -0.47 | 0.68 | | 0.09 | 1.03 | | -0.39 | 1.53 | | 0.00 | 0.00 | | 0.98 | 3.14 | |
| 4-8 | 7 | 3.75 | 3.04 | | 2.08 | 1.86 | | 5.41 | 4.08 | | 0.00 | 0.00 | | 0.53 | 1.21 | |
| 8-16 | 6 | 6.11 | 8.36 | | 2.54 | 10.88 | | 2.39 | 7.75 | | 0.55 | 0.82 | | 3.44 | 7.72 | |
| 16-32 | 5 | 7.93 | 10.72 | 5.87 | 1.92 | 13.33 | 7.97 | -1.85 | 0.00 | 2.84 | 19.39 | 13.32 | 15.27 | 0.82 | 3.91 | -0.77 |





| | | | | | | | | | | | | | | | | |
|---|---|---|---|---|---|---|---|---|---|---|---|---|---|---|---|---|
| **32-63** | 4 | 8.71 | | 9.96 | 6.13 | | 9.92 | 3.35 | | 7.46 | 20.51 | | 9.64 | 8.03 | | 9.44 |
| **63-127** | 3 | 20.30 | | 18.49 | 13.48 | | 14.01 | 6.81 | | 6.01 | 6.00 | | 4.73 | 9.76 | | 7.94 |

**Table 10: Change in post 2000 trend SE in the Full Model as a % difference of the Reference Model. Color coding is the same as introduced in Table 7.**

The MLO/Hilo location is close to the Tropical belt and therefore has different processes impacting stratospheric ozone variability as discussed in the previous section. We find that EqLat proxy can be added to the Full model in layer 6 and 5 (similar to other stations); however, above layer 6, EqLat or TP is not useful for interpretation of tropical ozone variability and therefore we believe the trend model in these layers should remain as it currently is used in Godin-Beekmann et al. (2022) analyses. The EqLat and TP are mildly correlated (-0.4) in the stratosphere, and therefore we decided against combining both of these proxies in the Full model. However, we also found that adding AO and QBO C/D proxies in layers 3, 4 and 5 improved the model fit and reduced the SE. These combined additional proxies are not correlated and reduce SE more than when using them separately.

The Full Model showed impacts on the SE in the upper stratosphere (above 8 hPa). The trend errors were reduced with the exception of Umkehr trends at 4–2 hPa over Boulder and Arosa/Davos where errors did not change. No changes in SE are found at MLO with additional proxies, thus the Full Model is kept the same as the Reference Model for this station in the upper stratosphere.

Similarly, in the middle stratosphere SE were mostly reduced after the Full Model was applied (except for slightly larger SE in trends derived from ozonesonde at OHP and from Umkehr at Boulder).

After applying the Full Model in the lower stratosphere, we still found high uncertainty due to higher ozone variability (natural variability), but SE were reduced. Arosa/Davos and MLO Umkehr and sonde trends changed after Full Model was used. Change in ozonesonde trends at HOH in layer 3 (125–63 hPa) goes from zero to positive and trend detection becomes highly confident (p-value <0.05). Umkehr trends at Arosa/Davos in layer 3 changed from negative to near zero but results have low certainty (p-value >0.1). Larger trend differences remain between ozonesonde and Umkehr at Lauder and OHP after the Full Model is applied.

| LOTUS Model Proxy Tests: (% difference of SE of Trend): overpass and zonal COH | | | | | | | | | | | |
|---|---|---|---|---|---|---|---|---|---|---|---|
| **Height** | **Umkehr** | **Arosa/Davos** | | **OHP** | | **Boulder** | | **MLO** | | **Lauder** | |
| **(hPa)** | **Layer** | **Overpass** | **Zonal** | **Overpass** | **Zonal** | **Overpass** | **Zonal** | **Overpass** | **Zonal** | **Overpass** | **Zonal** |
| **1-2** | 9 | 7.61 | 2.89 | 2.20 | 1.30 | 1.61 | 1.34 | NA | NA | 3.17 | 1.97 |
| **2-4** | 8 | 0.00 | 0.90 | 1.47 | 1.26 | 0.00 | 0.63 | NA | NA | 1.75 | 2.76 |
| **4-8** | 7 | 3.39 | 0.47 | 1.85 | 2.55 | 5.77 | 1.53 | NA | NA | 0.00 | 1.11 |
| **8-16** | 6 | 7.35 | 2.75 | 10.17 | 8.98 | 9.30 | 4.30 | 0.00 | 1.79 | 8.93 | 5.34 |
| **16-32** | 5 | 10.67 | 1.74 | 11.76 | 5.54 | 0.00 | 2.36 | 12.73 | 4.81 | 3.70 | -1.11 |

**Table 11: Change in Standard Error of Trend, as percent of Reference Model SE, for the COH overpass data and zonal data at the 5 ground stations. MLO Full Model in layers 9-7 is the same as the Reference Model (change is marked as NA).**



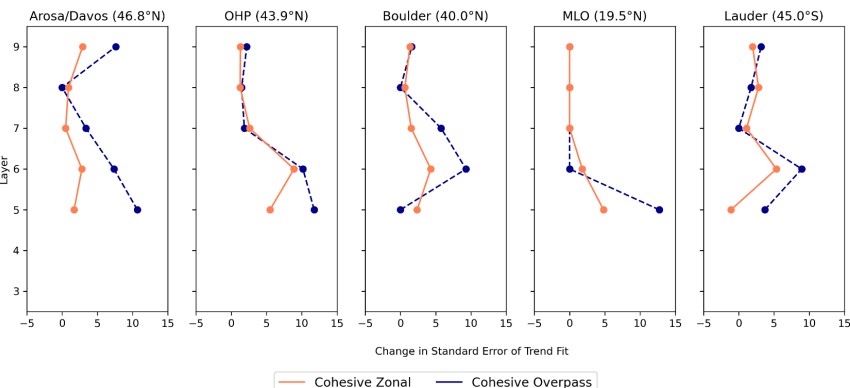

**Figure 6: Change in Standard Error of Trend, as percent of Reference Model SE, for the COH overpass data (blue) and COH zonal data (red) at the 5 ground stations.**

It is instructive to ponder if the addition of proxies that yield improvements via reduction of the standard error in the localized GB or overpass measurements also have the potential to improve uncertainties in the zonal data. To explore this Table 11 and Fig. 6 show the percent change in SE of the trend when adding the proxies for the Full model. Values are shown for both the COH overpass and the COH zonal data. In general, except when the improvement in the SE for the overpass COH is small (3% or less), addition of proxies has much less impact on the zonal results than on overpass results. This suggests that indeed the Reference LOTUS model is well tuned for zonal datasets, but can be improved with select addition of proxies for overpass or localized GB data.

**6.3 Impact of the Full Model on adjusted R2**

Table 12 shows the adjusted R2 for the Full Model. In the upper stratosphere, the Full Model increases the adjusted R2 above 8 hPa (except in Umkehr at 4–2 hPa). Over MLO there is no change because the Full Model is kept the same as the Reference Model for layers 7, 8 and 9.

| LOTUS Model Proxy Tests: (Adjusted of Model) | | | | | | | | | | | | | | | | |
|---|---|---|---|---|---|---|---|---|---|---|---|---|---|---|---|---|
| Height | Umkehr | Arosa/Davos | | | OHP | | | Boulder | | | MLO | | | Lauder | | |
| (hPa) | Layer | UMK | COH | SND | UMK | COH | SND | UMK | COH | SND | UMK | COH | SND | UMK | COH | SND |
| **1-2** | **9** | | **0.42** | | | **0.37** | | | **0.36** | | | 0.11 | | | **0.32** | |
| **2-4** | **8** | 0.23 | **0.39** | | 0.14 | **0.31** | | 0.17 | **0.39** | | 0.11 | **0.32** | | 0.18 | **0.34** | |
| **4-8** | **7** | **0.35** | **0.35** | | **0.31** | **0.41** | | 0.27 | **0.33** | | 0.26 | **0.32** | | 0.17 | 0.27 | |
| **8-16** | **6** | **0.31** | **0.35** | | **0.33** | **0.45** | | **0.33** | **0.40** | | **0.40** | **0.51** | | 0.25 | 0.23 | |
| **16-32** | **5** | **0.34** | **0.38** | 0.26 | 0.25 | **0.51** | **0.33** | **0.31** | **0.40** | 0.18 | **0.44** | **0.53** | **0.39** | **0.42** | **0.41** | 0.29 |
| **32-63** | **4** | 0.23 | | 0.25 | 0.29 | | **0.34** | 0.19 | | 0.18 | **0.42** | | **0.38** | **0.42** | | **0.31** |
| **63-127** | **3** | **0.31** | | **0.31** | **0.44** | | 0.26 | 0.22 | | 0.11 | 0.19 | | 0.24 | 0.25 | | 0.21 |

**Table 12: Adjusted R2 of the Full Model. Values of 0.30 and above are indicated in Bold as a threshold to indicate a satisfactory fit. Compare to Table 4 containing values for the Reference Model.**



In the middle stratosphere (32–8 hPa) adjusted R2 increases are found in all records (although smaller increases are
found in ozonesonde and Umkehr records at OHP, Boulder and Lauder at 32–64 hPa). At Arosa/Davos, Boulder and
Lauder the adjusted R2 in the COH and Umkehr trend models increase and continue to be very close in value. The
COH adjusted R2 is larger at OHP and MLO than in Umkehr and sonde records thus suggesting that overpass
conditions might have smoothed some natural variability observed in the GB records. In general, the adjusted R2 is
the largest at the 32–64 hPa level. This suggests that the Full Model shows an improvement for regional trend analyses
in the middle stratosphere.
Although Umkehr and sonde trend changes at MLO in the low stratosphere are within the SE and therefore can be
deemed not significant, the adjusted R2 is increased which suggests a better model fit in the Full Model. The adjusted
R2 increases in both Umkehr and ozonesonde data, while the largest increases are found in the Arosa/Davos, OHP
and MLO records.
In the lower stratosphere, the adjusted R2 remains low in both Umkehr and sonde records at Boulder (only TP is added
for the Full model). While the p-values at 63–32 hPa are significantly reduced (see discussion in the next section),
they still remain relatively high. These results suggest that additional research is needed to identify the best set of
proxies for Boulder records in the lower stratosphere. At Lauder, the ozonesonde record shows smaller adjusted R2
as compared to Umkehr partially due to low sampling biases.
It is valuable to further explore the impact of the Full Model on the adjusted R2 for the zonal and overpass COH data.
Fig. 7a shows the adjusted R2 for the Reference and Full Models at each of the 5 stations using the COH overpass
data. In all cases the Full Model improves the adjusted R2 except for MLO layers 7, 8 and 9 where the Full and
Reference Model are identical. The most significant improvements are seen by Umkehr at layers 3 to 7, COH overpass
at Layers 5, 6 and 7, and sonde layers 3–5. Figure 7b shows similar results using COH zonal data instead of overpass.
There is practically no further improvement in the adjusted R2 for the zonally averaged COH results (except for a
small increase for MLO layer 5). Comparison of results reveals that for OHP the implementation of the Full model
for the COH overpass data (Fig. 7a, dashed line) improves the adjusted R2 to values nearing that of the Reference
Model zonal data in layer 7 and below (Fig. 7b, solid line). For MLO and Lauder the use of the Full Model on the
COH overpass data improves the adjusted R2 over the Reference Model beyond the improvement seen in the COH
zonal results for layers 5 and 6. At Arosa/Davos and Boulder the implementation of the Full Model does not fully
reach the magnitude of the COH zonal adjusted R2.



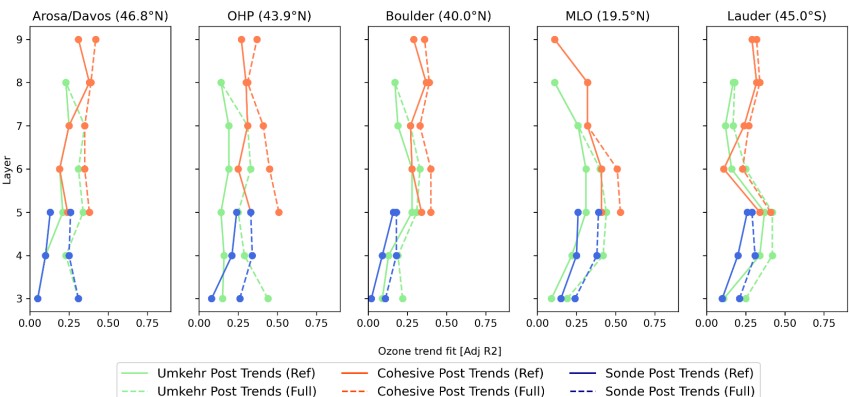

**Figure 7a: Adjusted R2 for the Full Model (dashed lines) and Reference Model (solid lines) at 5 stations. The COH data in**
**this figure is the overpass data at each station.**

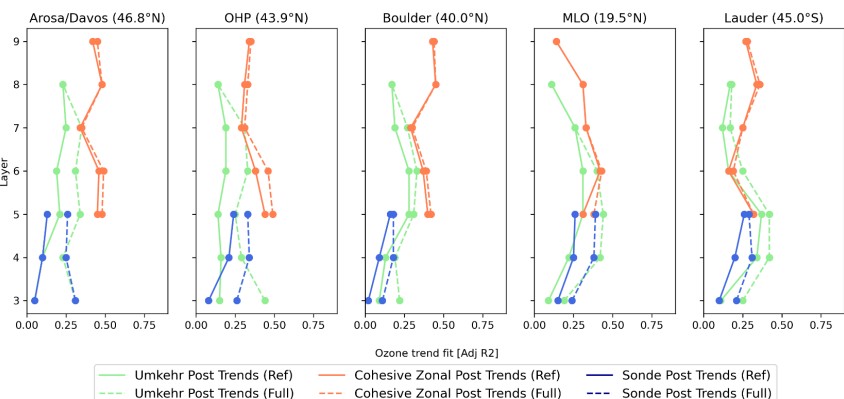


**Figure 7b: Adjusted R2 for the Full Model (dashed lines) and Reference Model (solid lines) at 5 stations. The COH data in**
**this figure is the zonal data for each station. The Umkehr and sonde lines are identical to those in Fig. 7a.**
**6.3 Examination of the p-values of the Full Model**
In the upper stratosphere (above 8 hPa), the confidence in Umkehr trends remained high (see Table 13) for most
stations except at Boulder (medium to low) and Lauder (very low, although some improvement was found). COH
trends confidence was very slightly degraded over Boulder at 1–2 hPa, but mostly has not changed.
In the middle stratosphere (between 32 and 8 hPa), p-values were significantly reduced in COH records. At 8–16 hPa
remained high, but at 16–32 hPa the confidence improved (continued) to high over Arosa/Davos and OHP (Boulder
and MLO). In case of Umkehr analyses in layer 8–16 hPa at Arosa/Davos, Boulder and MLO the confidence remained
high. However, at 16–32 hPa the Umkehr trend detection confidence was degraded over Arosa/Davos and Lauder.
For the ozonesonde record, the p-values remained low (<0.05) except at MLO where some improvement was found
after the Full Model was used, but the p-value remained high. It suggests that some instrumental records have either





high atmospheric or instrumental noise and therefore perhaps high certainty in trend detection cannot be achieved with
linear trend models. For near zero trends with high variability, the p-values are not a good criterion for trend
detectability.
In the lower stratosphere (between 125 and 32 hPa), analyses of p-values for the Full Model fit show significant
improvement for Umkehr trends at MLO between 63–32 hPa (while the p-value was increased at other stations at this
level). In addition, improvement in p-values was found for ozonesonde trends at all stations. Specifically, very low p-
values for the Full model were reached at Arosa/Davos (125–63 hPa), OHP (125–63 and 63–32 hPa), MLO (125–63
and 63–32 hPa), and Lauder (63–31 hPa).

| LOTUS Model Proxy Tests: (P Value of Model) | | | | | | | | | | | | | | | | |
|---|---|---|---|---|---|---|---|---|---|---|---|---|---|---|---|---|
| Height | Umkehr | Arosa/Davos | | | OHP | | | Boulder | | | MLO | | | Lauder | | |
| (hPa) | Layer | UMK | COH | SND | UMK | COH | SND | UMK | COH | SND | UMK | COH | SND | UMK | COH | SND |
| 1-2 | 9 | | 0.00 | | | 0.00 | | | 0.06 | | | 0.10 | | | 0.00 | |
| 2-4 | 8 | 0.00 | 0.00 | | 0.00 | 0.00 | | 0.06 | 0.00 | | 0.02 | 0.00 | | 0.41 | 0.00 | |
| 4-8 | 7 | 0.00 | 0.01 | | 0.02 | 0.00 | | 0.13 | 0.00 | | 0.00 | 0.00 | | 0.34 | 0.00 | |
| 8-16 | 6 | 0.01 | 0.17 | | 0.81 | 0.47 | | 0.02 | 0.83 | | 0.00 | 0.21 | | 0.92 | 0.71 | |
| 16-32 | 5 | 0.74 | 0.00 | 0.00 | 0.86 | 0.03 | 0.02 | 0.50 | 0.00 | 0.05 | 0.51 | 0.00 | 0.57 | 0.74 | 0.73 | 0.00 |
| 32-63 | 4 | 0.56 | | 0.34 | 0.67 | | 0.01 | 0.27 | | 0.15 | 0.09 | | 0.02 | 0.08 | | 0.01 |
| 63-127 | 3 | 0.66 | | 0.04 | 0.32 | | 0.01 | 0.31 | | 0.35 | 0.63 | | 0.02 | 0.02 | | 0.30 |

**Table 13: P Value of the Full Model. High certainty of trend detection is seen for values below .05 (green). Values between**
**.05 and 0.1 (yellow) medium certainty, between 0.1 and 0.33 (orange) low certainty of trend detection, and above 0.33 (red)**
**very low certainty or no evidence of trend detection.**
**7 Summary of the Full Model findings.**
We find that upper stratospheric trends in COH overpass and Umkehr records detect ozone recovery with high
confidence (p<0.05) above 8 hPa (with the exception of near-zero positive Umkehr trends over Lauder and Boulder).
We note the largest difference between Umkehr and COH trends (outside of the SE uncertainty) at Boulder, Mauna
Loa and Lauder.
Confidence for the middle stratosphere (between 32 and 8 hPa) trends vary between high, medium and low. Although
most of the trends are narrowly different from zero (especially when error bars are considered), there are some
differences in results across instrumental groups: trends in COH and sonde (except at OHP) between 32 and 16 hPa
tend to be small negative, while Umkehr trends are slightly positive. Some trends are statistically different from zero.
However, instrument-specific error bars often overlap and thus making differences in trends not significant.
Confidence in lower stratosphere trends is highly variable and even lower than in the middle stratosphere due to higher
ozone variability unaccounted for by Solar, QBO and ENSO proxies used in the Reference Model. However, high
confidence (p<0.05) is still found in ozonesonde trends at Arosa/Davos, OHP, MLO and Lauder (although not at all
layers). Umkehr trends in the lower stratosphere show lower confidence than ozonesonde trends (except at Lauder
and Arosa/Davos in the lowermost altitudes). The low confidence levels could be related to the near-zero trends
derived from Umkehr data, whereas ozonesonde trends are often different from zero lines. Also, we apply AK-



smoothing to the sondes to account for the wide AKs in the Umkehr retrieval. We tested the impacts of the AK on
ozonesonde trends (see Appendix A) and did not find any significant impacts. Most notably, ozonesonde and Umkehr
trends significantly disagree in the lower stratosphere at OHP and Lauder and therefore require further investigation.
The instrumental drifts and differences in Lauder trends are also discussed in Bjorkland et al. (2023 preprint) and are
consistent with our findings.

## 863    8 Conclusions

This paper is a follow up to Godin-Beekmann et al. (2022) with a focus on the GB record trend assessment. Therefore,
our trend analyses focus on the questions:

1)   Do proxies for evaluating trends of GB stations need to be different from those of the optimized set for zonal
data?
2)   Are station records representative of the small geophysical region or semi-global changes?
3)   Do uncertainties of the zonal averaged trends improve with additional proxies?


The Full Model developed in this paper for station and overpass data adds proxies to the LOTUS models of Godin-
Beckmann (2022). Our trend analysis of stratospheric ozone records from the Umkehr, ozonesonde and COH station
overpass data at 5 geographical regions using the Full Model (LOTUS v 0.8.0) show similar trends to those published
in Godin-Beekmann et al. (2022) paper. We analyze trends for instrumental records converted to 7 Umkehr layers that
represent ozone changes in the upper, middle and lower stratosphere over NH and SH middle latitudes and over high
tropics of the NH. We also analyze GB station records at Arosa/Davos, Hohenpeißenberg and OHP separately in
contrast to the "European regional" trend analyses presented in Godin-Beekmann et al. (2022) and included COH
overpass records for comparisons with the GB records. Our analyses include evaluation of the adjusted R2 (aka
goodness of the model fit), standard error and p-values.
We also investigate differences between satellite trends as detected in the records sampled for individual geographical
locations (spatial and temporal overpass criteria) versus zonal average datasets. We find that COH overpass ozone
records capture ozone variability of the ground-based station records (Umkehr and sonde) better than COH zonal data.
We do not find that the COH zonal record is improved by using EqLat instead of geometric latitude to construct the
dataset (see Appendix C), but EqLat can be an important additional proxy at some levels for GB data. To determine
the improvement to the model fit we use the Standard Error and adjusted R2 for the Full and Reference model fit.
Using the Reference model for the zonal mean COH data we find slightly better adjusted R2 than for the COH overpass
data fit over the Northern middle latitude stations. This is expected as much of the variability of the overpass time
series is reduced in the zonal average data. Therefore, we also explore the impact of additional predictors in the trend
model fit applied to the more variable GB and satellite COH overpass data to determine if that will reduce the SE and
improve the adjusted R2. We also apply the Full model to the zonally averaged data to assess the benefits of additional
proxies to further reduce trend uncertainties.
We find that adding predictors (with few exceptions) does not change the trends but often reduces SEs and increases
the adjusted R2 (with the exception of the upper stratospheric ozone trends at MLO). We also find that the p-values





are useful for interpretation of improvements of the model fit in the data, although improvements in the SE do not
always result in improved confidence in derived trends, especially when the trends are close to zero. In these cases we
conclude that either longer records are needed to discern trend information outside of the atmospheric noise or further
research into the inconsistencies between instrumental records and homogenization procedures is required. We also
find the small changes in trends in the lower stratosphere and improvements in the model fit after additional proxies
are used. However, the sampling tests indicate that trends can depend on the temporal selection of the records when
AK are used to smooth ozonesonde high resolution profiles (see discussion in Appendix D).
This paper concludes that additional proxies bring improvements to trend detectability in case of GB and gridded
satellite data analyses and better agreement is achieved between satellite overpass and GB trends. We also find that
zonally averaged and gridded satellite records produce comparable trends over the studied middle latitudes and
subtropical regions. Therefore, the GB trends are representative of the stratospheric ozone changes over the semi-
global area. Finally, zonally averaged data do not benefit from addition of proxies beyond what LOTUS model uses
for global trend detection whereas the uncertainties in GB and gridded trends are significantly reduced and sometimes
(Boulder, MLO, Lauder) become comparable to the uncertainties of the zonally averaged trends in the upper and
middle stratosphere. Based on analyses presented in this paper we strongly recommend using additional proxies for
trend analyses of GB and gridded satellite stratospheric ozone records. Additional proxies should be selected based
on the latitude and altitude of the observational ozone record to adequately represent stratospheric transport and mixing
processes impacting interannual and seasonal ozone variability.
**Appendices**
**Appendix A: AK Smoothing for ozonesondes**
Ozonesonde profiles have high vertical resolution (purple line in Fig. A1) in comparison to the Umkehr (green solid
line) or COH (orange dashed line) ozone profiles. Each Umkehr layer is referenced to the atmospheric pressure at the
bottom of the layer, which is constructed using half of the pressure in the layer below. Averaging Kernels (AK) as
shown in Fig. A1, panel b, define the granularity of the Umkehr vertical grid. In order to compare trends from three
instrumental records in the same vertical system, we convert the ozonesonde and COH profiles to the Umkehr layers
and DU. The COH overpass data is in units of DU, but on different layers than the defined Umkehr layers, so only
vertical grid modification is required. The sonde profiles (purple thin line) are in units of partial pressure and are first
converted to DU, then converted to the Umkehr grid (blue solid line in panel a). Conversion to the Umkehr grid can
be done either by interpolation, or by AK smoothing. The equation describing the process of applying AK smoothing
is

$$Ozone_{smoothed}(i) = \sum_j \{AK_{i,j} * Ozone_{true}(j) - Ozone_{apriori}(j)\} + Ozone_{apriori}(i)$$


where AK is the Averaging Kernel for layer i, $Ozone_{smoothed}$ is the smoothed ozone result, $Ozone_{true}$ is the
ozonesonde profile, and $Ozone_{apriori}$ is the Umkehr a priori (climatological) profile. The AK for each Umkehr layer



is used as a weighting function applied to the ozonesonde profile (Ozone$_{true}$) prior to the integration which simulates
the Umkehr optimal estimation method used for estimating the ozone content in the targeted layer (Rodgers, 2000).

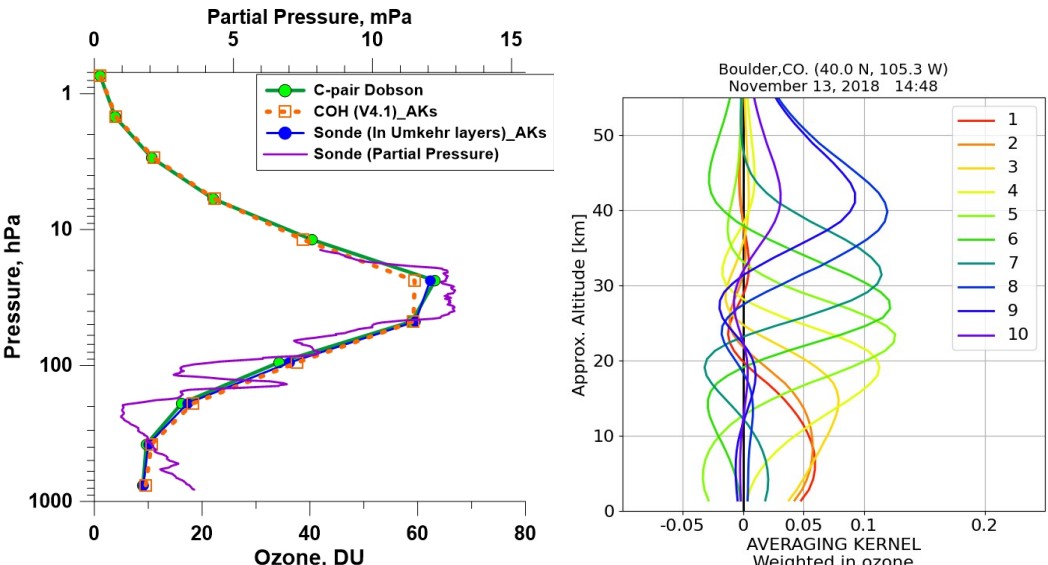


**Figure A1: a) An example of ozone observations over the Boulder, CO station. The purple line is 100-m averaged ozone partial pressure (hPa) vertical profile measured by sonde on 13 November, 2018 . The green line with solid circles is the ozone profile derived from Dobson Umkehr observations on the same day. The blue line with blue dots is the ozonesonde profile converted to the Umkehr layers and smoothed with the Umkehr AK. The orange dashed line with open squares is the COH ozone profile observed over Boulder on the same day and interpolated to the Umkehr layer vertical grid. b) The Umkehr AK for the ozone profile derived from observations in Boulder on 13 November, 2018. Each line represents the smoothing function for one of 10 Umkehr layers (see color legend).**

Although the ozonesonde measurement typically reaches altitudes between 32 and 10 hPa, the balloon often bursts
before reaching the top of layer 6 (16 hPa), therefore only partially covering the ozone content in that layer. We also
note that Umkehr AKs are relatively wide and therefore will incorporate (weight in) ozone variability from the layer
above and layer below of the targeted Umkehr layer. (See layer 6, green line in Fig. A1, panel b.) Therefore, there
are two sources of error in ozonesonde comparisons with Umkehr ozone in layer 6: a) burst level for ozonesonde does
not reach the top of the layer 6, thus the integrated ozone is smaller than expected. b) the Umkehr AK for layer 6 is
relatively wide and therefore the Umkehr layer partially contains information from above the burst altitude of the
ozonesonde, thus making smoothed ozonesonde concentration lower than expected. In order to avoid these errors, we
only show ozonesonde results up to layer 5.
Similarly, we explored smoothing COH profiles with Umkehr AKs. Figure A2 demonstrates the time series of the
COH ozone over the Mauna Loa station. The trend model was fitted to the COH record with and without AK applied.
The reference trend model included proxies and trends. To focus on ozone variability that contributes to the trends we
subtracted the modeled ozone variability from the COH data and then added the trend component back. The COH
record residuals in Fig. A2 are shown in Umkehr layers where COH is either smoothed with AK (red lines) or not
(green lines). We notice that the AK-smoothing of the COH profile in layer 9 does not have a lot of independent





information from layer 8.  In this example it clearly shows that the trends in layer 8 are embedded in the COH layer 9
ozone time series, which was confirmed when we compared trends derived from the AK-smoothed COH in layers 8
and 9. In case of the integrated COH ozone record, the trends in layers 8 and 9 differed.  In order to avoid biasing the
COH trends at layer 9 we decided to not apply Umkehr AKs for COH smoothing and only use COH profiles
interpolated into the Umkehr layers.  This result makes sense since COH overpass data are derived from UV
backscatter radiances also using an Optimal Estimation technique.  COH overpass data has a comparable vertical
resolution to Umkehr, simply with different layer definitions.  Interpolation makes the most sense for rendering COH
data in the Umkehr vertical coordinate system.

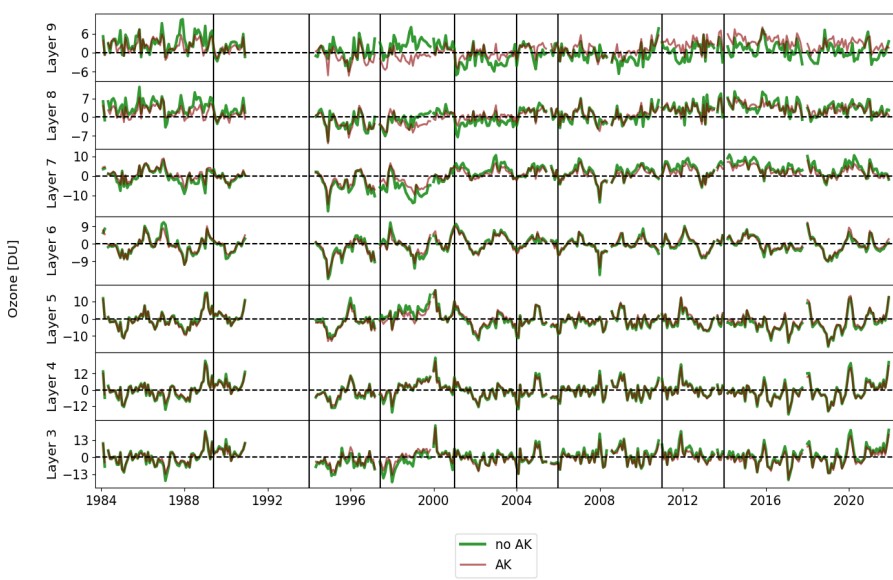


**Figure A2: Modified residuals (seasonal cycle, Solar, QBO, and ENSO are removed, but trend is retained) of COH overpass data at Mauna Loa (20N, 156W). Red: AK smoothed to Umkehr layers; Green: Interpolated to Umkehr layers. Vertical lines show the dates of satellite records in COH. The largest impact of the AK is seen between 1997 and 2001 where two curves separate in layers 7, 8 and 9, and also after 2001 in layer 9.**

Figure A3 demonstrates time series of monthly mean ground-based records the lower stratosphere at 5 stations. The
Umkehr data (blue) are compared with the ozonesonde anomalies either interpolated to the Umkehr layer 3 (green),
or ozonesonde profiles matched with Umkehr profiles in time and smoothed using the Umkehr averaging kernels
(crimson). All three datasets have been deseasonalized using their respective climatological (using 1998-2008
climatology) average monthly mean ozone. The application of the Averaging Kernels has the effect of smoothing the
temporal variability.

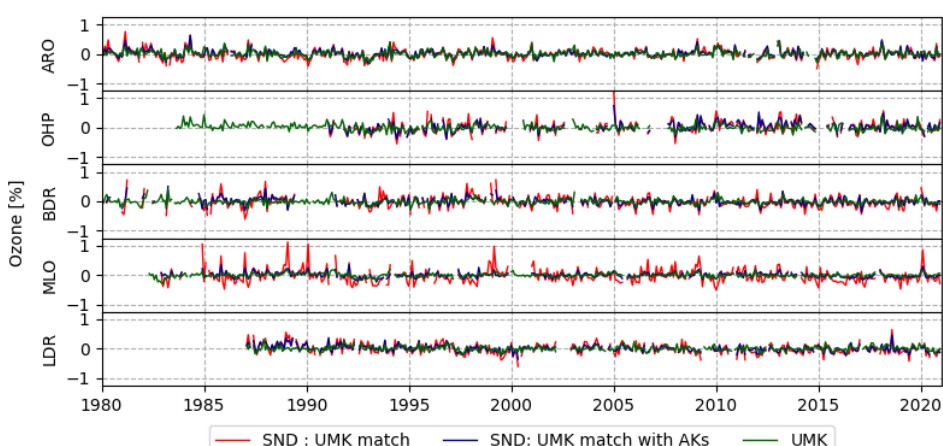


**Figure A3: Time series of monthly averaged and de-seasonalized (in %) ozone anomalies of Umkehr (green) and ozonesondes records are compared at 5 ground-based stations. Ozonesonde data are either calculated using only profiles that are interpolated in Umkehr layer 3 (blue) or matched with Umkehr profile in time and smoothed with the Umkehr averaging kernels (crimson).**

**Appendix B: COH using OMPS v3r2 vs OMPS v4r1**

OMPS SNPP v4r1 uses updated SDRs as input which incorporate unified and consistent calibration algorithms removing artificial jumps caused by operational changes, instrument anomalies, or contamination for anomaly views of the environment or spacecraft. Also included are new interpolated band-passes, and updated soft calibration based on the new input SDR's.

Differences between the v3r2 and v4r1 versions of the resulting COH dataset are typically less than 1 percent (Fig. A4 and A5). Small seasonal variation is apparent at all levels. Larger differences are visible in 2020 when the soft calibration for v3r2 is extended beyond its period of relevance. Figure A6 shows the drift between the two versions. Drift between the datasets is less than +- 1% at all levels. This is a reasonable estimate of the resulting expected trend difference in using the newest COH version as compared to the v3r2 results used in Godin-Beekmann (2022).



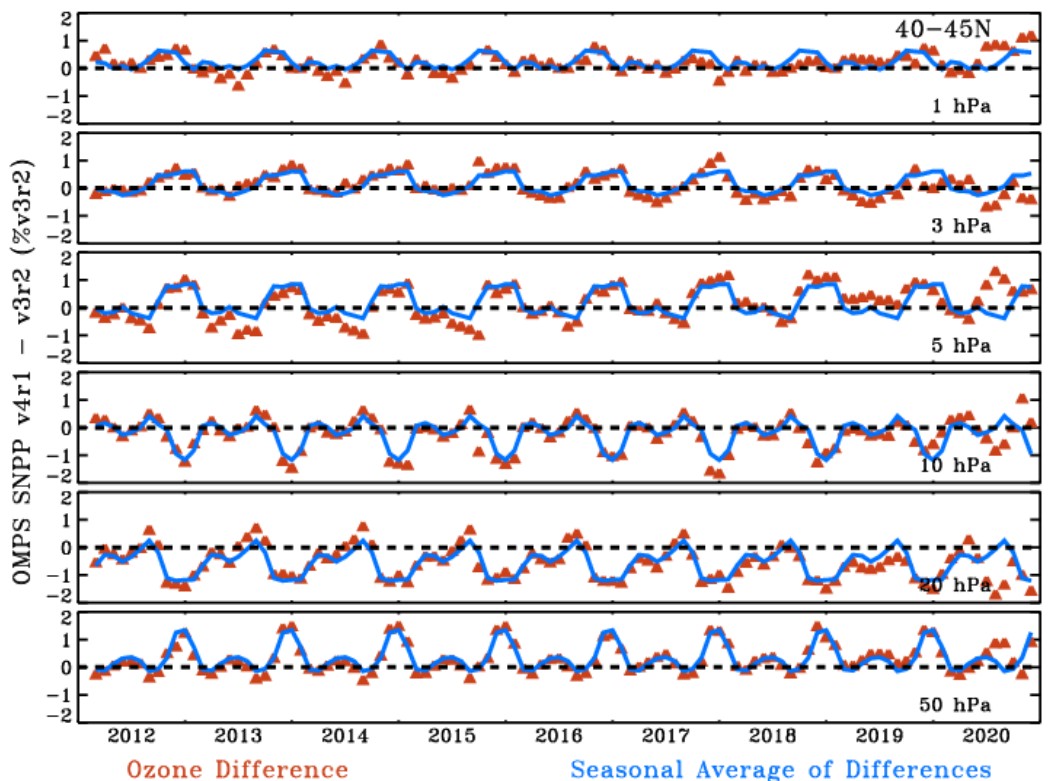



**Figure A4: Differences in the COH monthly average zonal product as generated from SNPP v4r1 and v3r2 processing.**
**Also shown is the annual cycle in this difference as depicted by the average over all years for each month. Exhibited at 40-**
**45N is a less than 2% difference with an annual cycle. A somewhat different pattern is seen in 2020 where the soft**
**calibration for v3r2 is extended beyond its period of relevance.**



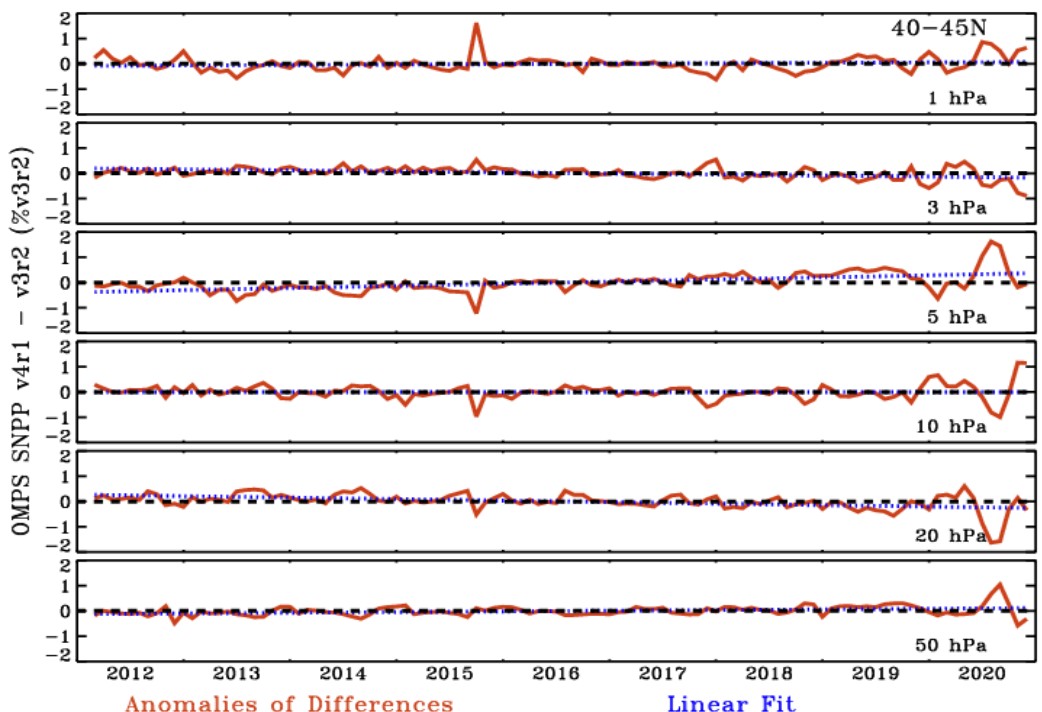

**Figure A5:** Anomalies of the differences in version (v4r1 vs v3r2) in the COH monthly average zonal product at 40-45N. Anomalies are enhanced in 2020. Also shown as a blue dotted line is a linear least square fit to the anomalies representing the drift between the two versions.

10000



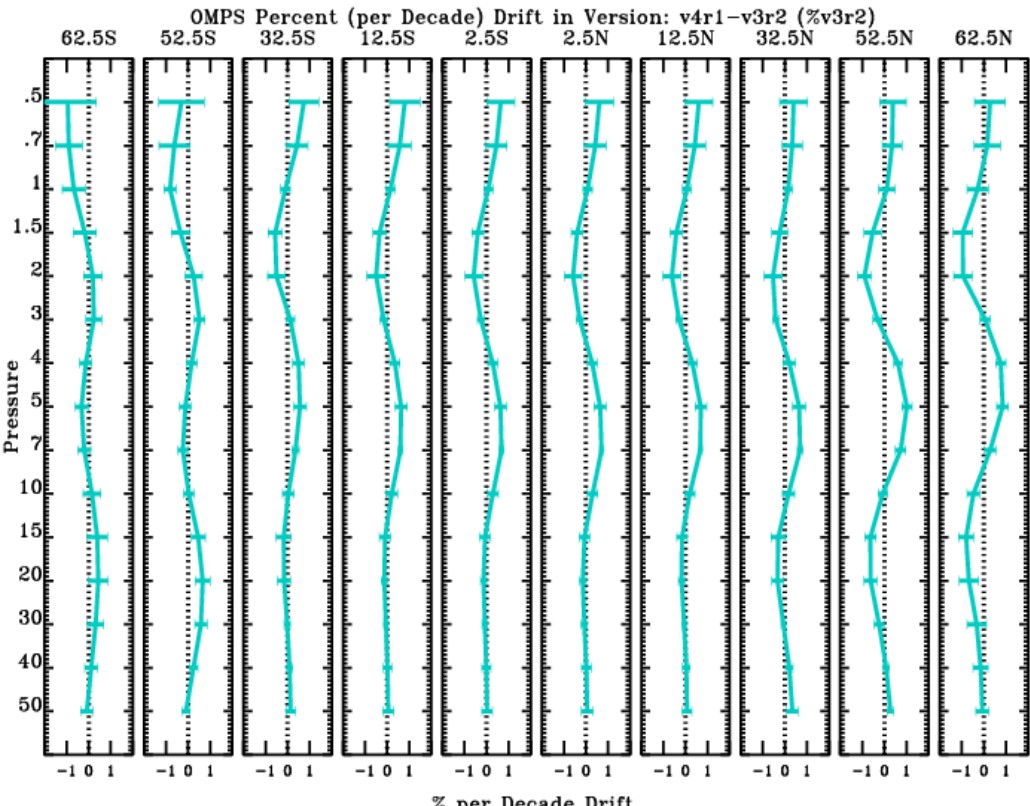

**Figure A6 shows the drift between the two versions (v4r1 vs v3r2) as function of pressure level at 10 latitudes.**

**Appendix C: Impact of using equivalent latitude in generation of the COH product**

The COH overpass data used in this paper collects all profiles during the day within a latitude and longitude box of +/- 2 degrees by +/- 20 degrees, then generates a 1/distance averaged value for the station. The box is based on geometric latitude and longitude. With 15 orbits per day, the chosen box size guarantees 2 to 4 possible profiles within the box depending on whether the orbit overpasses or straddles the site as shown in Fig. A7. Also shown is a scenario when the equivalent latitude (EqLat) near the site is particularly non-zonal. In such cases the profiles selected using a geometric coordinate box will select SBUV profiles from an Eq Lat that is different from that of the measurement station.



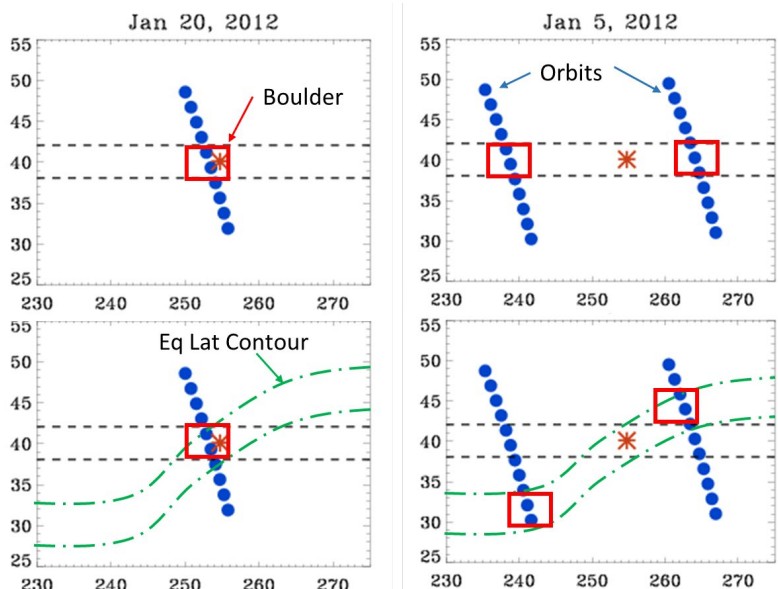

**Figure A7: Shows orbits of SNPP and positions of OMPS NP ozone profiles on January 20, 2012 and January 5, 2012. The second row displays a possible EqLat contour overlaid.**

It is informative to create an overpass product using boxes based on EqLat and determine the impact on the data. Since EqLat is layer dependent, the included profiles must be selected independently for each layer. Figure A8 shows COH overpass data for Boulder using geometric coordinates, EqLat based coordinates, and the associated Umkehr data. Color coding shows the EqLat at Boulder for each measurement day with dark blue and yellow indicating days with extreme variation from 40N.






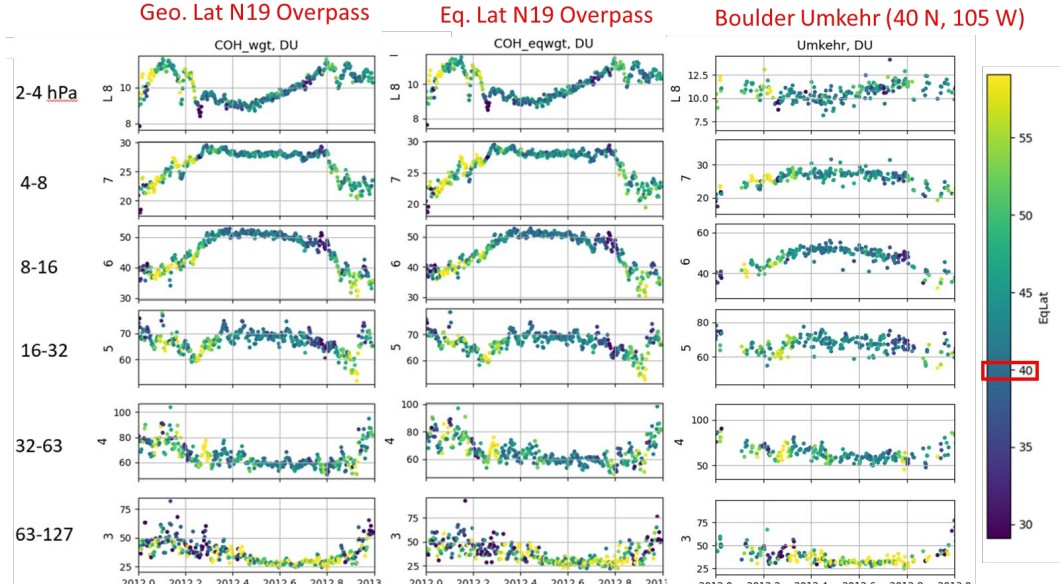

**Figure A8: COH overpass data generated with geometric coordinates, EqLat based coordinates, and the associated Umkehr**
**dataset at Boulder for 2012. Data points are color coded for the EqLat at the measurement site. Boulder is at 40 N.**
Variation in EqLat is most apparent in Winter months and transitional Fall and Spring, less so in Summer. Yet the
value of the COH ozone is not dramatically altered in the time series. Figure A9 shows correlation plots of the COH
overpass to Umkehr for the data at layer 7 (4-8 hPa). The pattern of the scatter and the value of the correlation
coefficient are not substantially altered for overpass determination using geometric latitude (left) and EqLat (right).
Figure A10 shows the vertical distribution of the Correlation coefficient and the RMS Difference for the two COH
datasets vs Umkehr. These two metrics are minimally impacted for this sample year in the layers where COH is valid.

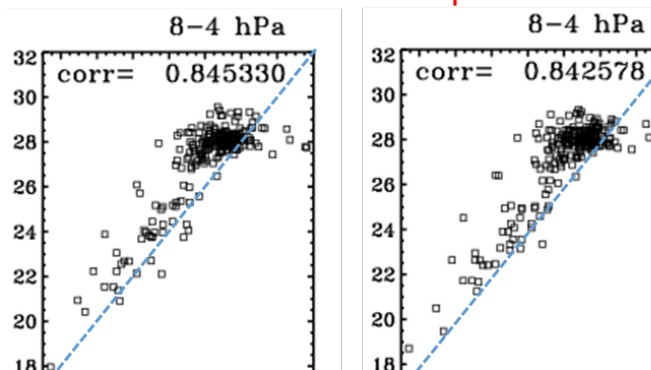




**Figure A9:  Correlation between Umkehr and COH overpass using Geometric Latitude (left) and EqLat (right) to select included profiles for layer 7 (4-8 hPa).**

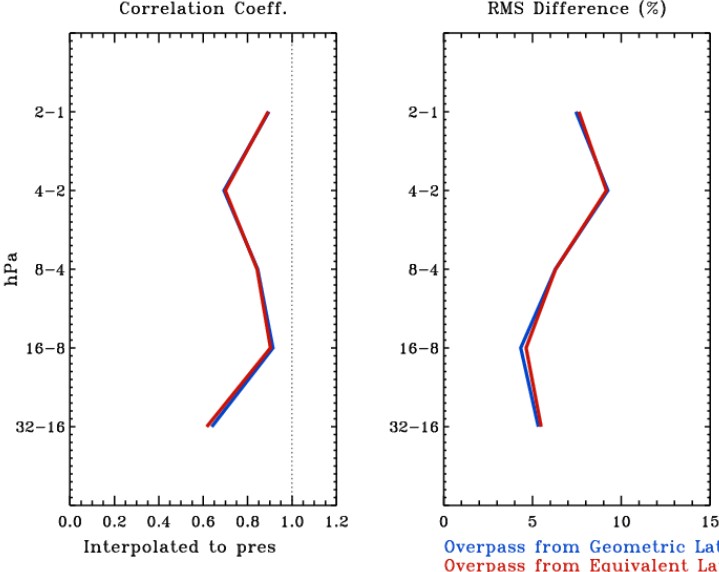

**Figure A10: Profiles of Correlation coefficients and RMS differences between COH overpass data at Boulder for 2012 using Geometric Latitude (blue) and EqLat (red) to select data points included in the average.**

The use of geometric latitude appears to be sufficient in the choice of included data points in the overpass COH product at the layers used in this paper.  Likely this is a ramification of the smooth horizontal resolution of the satellite product.

**Appendix D: Temporal Sampling and Impact on Trends**

This paper compares trends for three instrument types each with differing measurement frequency.  From each set of measurements a monthly average is constructed.  See the data files at https://gml.noaa.gov/aftp/ozwv/Publications/2023_Umkehr_Ozone_Trends_Paper/ for the data and the number of data points in each monthly average with the sampling variations.  Umkehr measures once or twice per day depending on cloud interference with the measurement.   At Arosa/Davos and Lauder, Umkehr measurements are sparser than the other GB stations, often less than 10 per month. At Boulder beginning in 1983 measurements number 20 or more per month.  At OHP the Umkehr record begins in 1983 with a strong 20 or more measurements per month.  From 1999 to 2016, however, measurements per month are often less than 15 per month. The most Umkehr measurements at MLO are the most abundant, especially after 1985 measuring multiple times in a day, resulting in 50-70 data points contributing to the monthly average. The COH overpass dataset is typically available once per day at each station with occasional misses, contributing usually 27-30 data points per month. Since Umkehr can measure multiple times per day, the COH data matched to Umkehr can contain more profiles in the monthly average than the original full COH data, since the COH overpass data will appear twice in the monthly average, once per each Umkehr measurement.





This occurs often at MLO. Ozonesonde launches are typically one to three times per week depending on the station.
At Arosa/Davos, sonde measurements are typically about 15 per month. Sonde measurements at the other stations
usually have approximately 5 measurements per month, with some periods of up to 10 per month. As with COH
overpass measurements, the sonde dataset matched to Umkehr can have more contributions to the monthly average
resulting from dates with more than one Umkehr measurement, resulting in multiple sonde matches.
The trend results in this paper use all available Umkehr data to generate the monthly means. The COH and sonde data
are matched to Umkehr to use the Umkehr temporal sampling for COH, and to be able to use the Umkehr averaging
kernels for sonde. It is important to determine how the temporal sampling within the monthly mean data may impact
trend results. To aid this understanding, we create three subsets of Umkehr data each with different temporal sampling
and create the corresponding monthly mean: 1) all observations in Umkehr record; 2) Umkehr matched to the COH
dataset; and 3) Umkehr matched to the sonde dataset. In this way we use the same data, but only vary the temporal
sampling. Since the COH is measured every day, except in the rare case that the satellite data is missing due to
instrument issues, sampling 1 and 2 should provide nearly identical results. We expect a strong change in the monthly
mean and resulting trends for Umkehr record when it is matched with infrequent sampling of ozonesonde profiles
(especially in Boulder, Hilo and Lauder).
Figure A11 summarizes the results. Each line in Fig. A11 is trend derived from Umkehr data, but with sampling of all
data, data matched to COH dates, and data matched to sonde dates. In general, the differences are within the envelope
of trend uncertainty (+/- 2 std errors). As expected, the trends and standard errors for all (green) and COH-matched
subsampled (orange) Umkehr records are nearly the same. The largest differences in all Umkehr and COH matched
Umkehr lines are apparent at OHP. We have determined that this arises from occasional months when there is a short
satellite outage coupled with sparse Umkehr observations at the station. However, trends derived from sonde-matched
Umkehr data (blue) show deviations from other observations. This is especially clear at Arosa/Davos in the upper
stratosphere (~2-3 % above 10 hPa). But since this is above the measurement capability of the ozonesonde, this will
not impact the ozonesonde trend results at Arosa/Davos. At Lauder the most significant differences are seen in layer
3 (2.5%),but unfortunately not in the direction to explain sonde differences in the Lauder trend curves as compared to
Umkehr. Smaller differences are seen at other layers (very small, less than 1 %, differences in layers 6 and 4). At
OHP small differences of less than 1 % are seen between 50 and 10 hPa, well within error estimates.



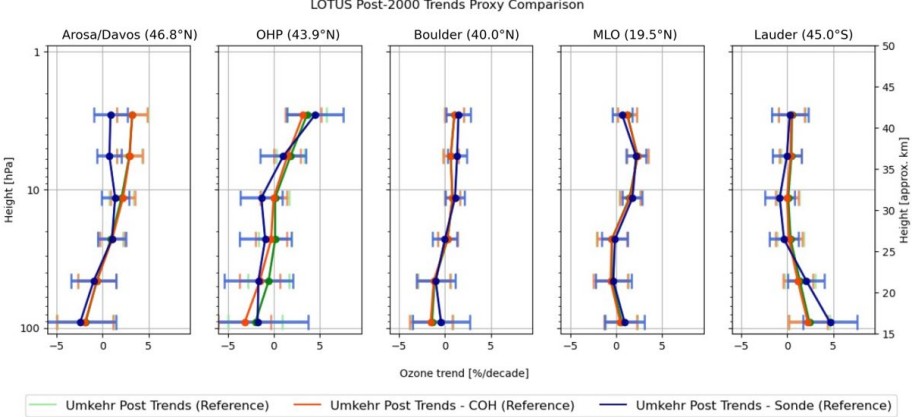


**Figure A11: Trend results for the Reference Model using Umkehr data mimicking the temporal sampling of COH and sonde. Green is all available Umkehr data; orange is Umkehr data matched to COH measurements dates; blue is Umkehr data matched to sonde measurement dates.**

Figure A12 further explores sampling differences by examination of trends of COH data using the full COH dataset, and data sampled to the Umkehr dates in generation of the monthly mean datasets. As with Fig A11. the trend lines are nearly identical at all stations except OHP. At OHP in the early 2000's there are significantly fewer COH points matched to Umkehr because of the drop in Umkehr measurements. This likely impacts the post-2000 trend estimate. The differences remain below 2%, and are within the error estimate of the trends. In summary, the sampling biases between COH overpass and Umkehr data cannot explain the difference in the derived trends (see Fig. 3, most notable in layers 7 and 8 at Boulder and Lauder).


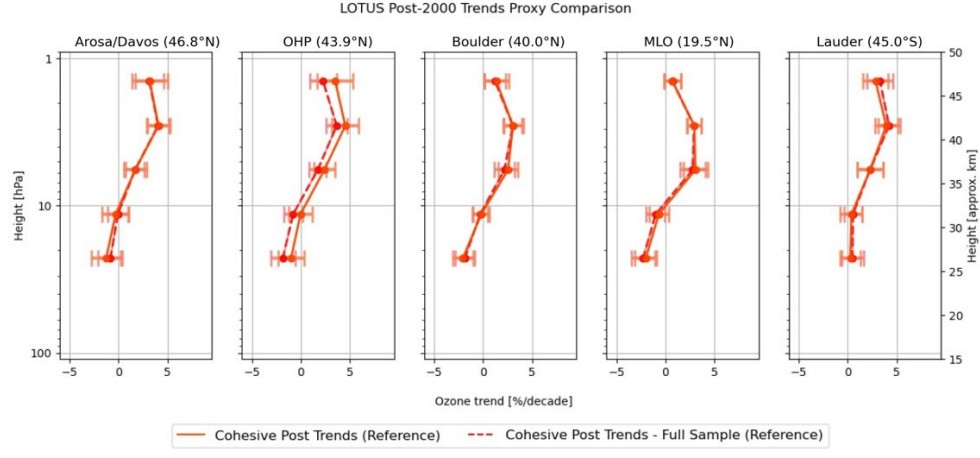


**Figure A12: Trend results for the Reference Model exploring variations in sampling of the COH data. Solid orange is COH data matching Umkehr sampling; dotted orange is all available COH data.**




Figure A13 explores the impact on trends from sampling differences of the sonde data. Shown are trends with all
sonde data, and trends with Umkehr matched data. In this figure only, the sonde data is not AK smoothed since the
Umkehr AK are only available on dates when there is an Umkehr measurement. So shown here are trends from sonde
data integrated to the Umkehr levels. As with Fig. A11 the only visible impact is seen at OHP and Lauder, though
both are within error estimates. At Lauder the trends remain negative for both samplings, but sonde sampled to
Umkehr moves closer to the zero line. At OHP the sonde trends are positive, but sonde sampled to Umkehr moves
slightly closer to zero. The sampling impact on trends for both OHP and Lauder are likely due to the reduced number
of Umkehr data at these sites.

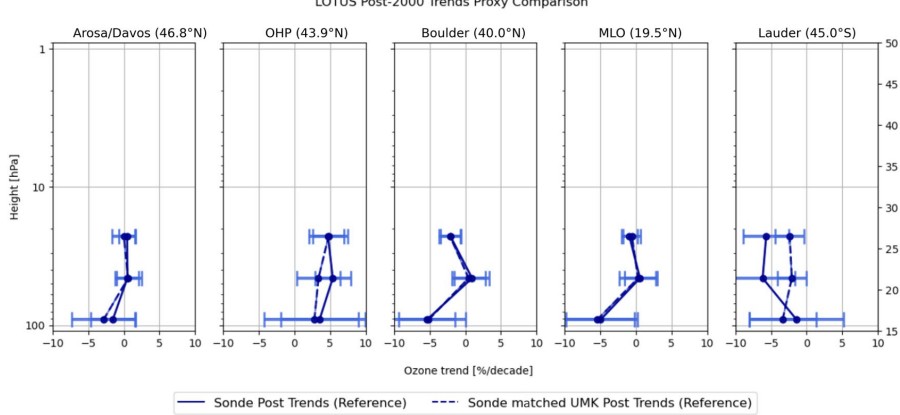


**Figure A13: Trend results for Reference model exploring sampline of the sonde data. Solid blue is all sonde data; dashed**
**is Umkehr matched sonde data.**
Figure A14 explores the impact of sampling on the adjusted R2 using the COH overpass data. Shown are the adjusted
R2 for all available COH overpass data, and the same using only COH overpass with matches to the Umkehr data.
For Arosa/Davos, OHP and Lauder the differences are small. For Boulder and MLO at some layers (Boulder, layers
6,7; MLO layers 6,9), the impact is more apparent with the Full COH exhibiting higher adjusted R2 at these stations.





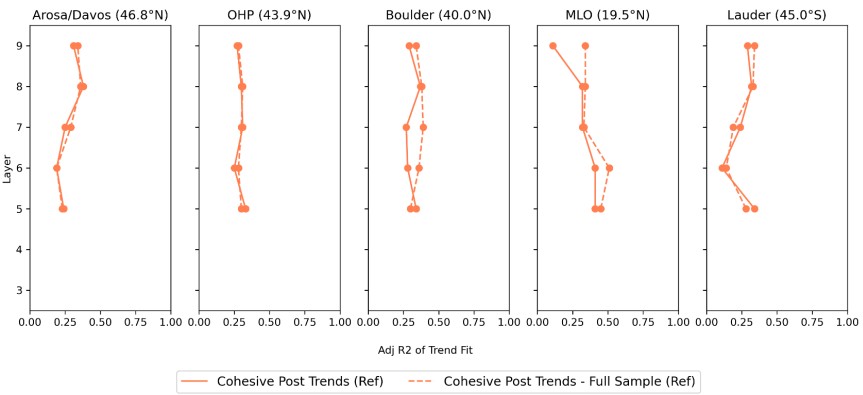


**Figure A14: Adjusted R2 for the Reference Model exploring variations in sampling of the COH data. Solid orange is COH data matching Umkehr sampling; dotted orange is all available COH data.**

**Appendix E: Decision process for the Full Model**

The LOTUS styled Reference Model is developed and optimized for zonal average datasets. The Extended Model tests the addition of single predictors to see if fit statistics can be improved for GB and overpass datasets. For Tropospheric Pressure (TP), improvements are consistent among layers and among instrument types. The addition of EqLat also yields consistent results for instrument types and at most stations, though not Mauna Loa. Addition of other predictors gives mixed results depending on level and station. The potential for improving confidence in trend results exists by combining predictors using different choices depending on layer and station. We choose additional predictor combinations with consideration of three criteria: 1) combined predictors should not have a high correlation with each other (usually .2 or less); 2) predictors should reduce the SE of the trend consistently for all instrument types; 3) addition of the predictor should not greatly reduce the adjusted R2 of the model fit, but preferentially increase it. As seen in Tables 7e and 7f the NAO and the EHF predictors do not make a significant improvement when added to the Reference Model, so we do not include either in the Full Model.

**Mixed Model:**

We have noted a high correlation between the TP and EqLat predictors at all levels especially for Boulder, Mauna Loa and Lauder with correlation adjusted R2 of .4 to .7, and somewhat less correlated at Arosa/Davos and OHP with adjusted R2 of .2 and .3. Subsequently, we choose to not use these two predictors together (at the same station/layer combination). The addition of TP at all stations for layers 3 and 4 uniformly decreases the standard errors at all stations for both Umkehr and sonde. The addition of EqLat (with the exception of Umkehr at Boulder, level 5) almost uniformly decreases the standard errors at all stations for layers 5 and 6. There is additional reduction in the SE for layers 7 to 9 for all stations except at Mauna Loa. Thus, we choose TP and EqLat as additional predictors at these layers. QBO C and D, have significant impact in decreasing the SE in layers 4 and 5 for both Umkehr and sonde, and layer 3 for sonde with only a small degradation for Umkehr. QBO-CD shows an improvement in layer 8 at OHP, both





COH and Umkehr, and Arosa/Davos and Boulder for COH only. We have tested adding both QBO and EqLat for layer 8 at these 3 stations. For Umkehr measurements, there is no improvement beyond EqLat only with QBO-CD also included. For COH there is additional improvement, but not to the extent of QBO-CD alone. Since the improvement is limited to one layer, and for only COH, we choose to only add the additional QBO-CD for the tropical MLO. Table A1 shows the resulting combination of additional predictors for this Mixed Model.

| LOTUS Mixed Model | | | | | |
|---|---|---|---|---|---|
| **Layer** | **Arosa/Davos** | **OHP** | **Boulder** | **MLO** | **Lauder** |
| **9** | EqLat | EqLat | EqLat | Ref | EqLat |
| **8** | EqLat | EqLat | EqLat | Ref | EqLat |
| **7** | EqLat | EqLat | EqLat | Ref | EqLat |
| **6** | EqLat | EqLat | EqLat | EqLat | EqLat |
| **5** | EqLat | EqLat | EqLat | EqLat, QBO CD | EqLat |
| **4** | TP | TP | TP | TP, QBO CD | TP |
| **3** | TP | TP | TP | TP, QBO CD | TP |

**Table A1: Details of additional predictor combinations for each level and station in the Mixed Model**

The resulting change in SE from the Reference Model is shown in Table A2. For most stations/layers this is simply a composite of the values from the single EqLat or TP Extended Model results. There remain a few instrument/station/layers where the SE is slightly increased - OHP sonde Layer 5, Arosa/Davos Umkehr layer 8 and Boulder Umkehr layer 8, but these are negligible. At Boulder Layer 5 Umkehr the increase in SE is somewhat more at 1.85% difference, but this is still small enough to not be of great concern. For Mauna Loa at layers 3,4 and 5 the model is rerun adding two predictors together and the results are new. Indeed in these cases the SE is improved beyond the single predictor results of either QBO alone, or TP or EqLat alone with the exception of Sonde layer 5 where the change in SE is just slightly degraded from QBO alone (13.42% vs 13.69% reduction in SE).

| LOTUS Model Proxy Tests: (% Difference in Std. Error of Model) | | | | | | | | | | | | | | | | |
|---|---|---|---|---|---|---|---|---|---|---|---|---|---|---|---|---|
| **Height** | **Umkehr** | **Arosa/Davos** | | | **OHP** | | | **Boulder** | | | **MLO** | | | **Lauder** | | |
| **(hPa)** | **Layer** | UMK | COH | SND | UMK | COH | SND | UMK | COH | SND | UMK | COH | SND | UMK | COH | SND |
| **1-2** | **9** | | 8.35 | | | 2.74 | | | 1.94 | | | 0.00 | | | 2.85 | |
| **2-4** | **8** | -0.47 | 0.68 | | 0.09 | 1.03 | | -0.39 | 1.53 | | 0.00 | 0.00 | | 0.98 | 3.14 | |
| **4-8** | **7** | 3.75 | 3.04 | | 2.08 | 1.86 | | 5.41 | 4.08 | | 0.00 | 0.00 | | 0.53 | 1.21 | |
| **8-16** | **6** | 6.11 | 8.36 | | 2.54 | 10.88 | | 2.39 | 7.75 | | 0.55 | 0.82 | | 3.44 | 7.72 | |
| **16-32** | **5** | 7.93 | 10.72 | 5.87 | 1.92 | 13.33 | -0.56 | -1.85 | 0.00 | 0.90 | 15.21 | 11.13 | 13.42 | 0.82 | 3.91 | 0.00 |
| **32-63** | **4** | 6.60 | | 6.07 | 5.87 | | 7.04 | 3.35 | | 2.97 | 13.15 | | 9.70 | 8.03 | | 4.13 |



| 63-127 | 3 | 12.80 | | 10.17 | 12.80 | | 6.91 | 6.81 | | 2.46 | 3.81 | | 4.19 | 9.76 | | 4.41 |

**Table A2:  Change in the SE of the trend using the Mixed Model.**

Table A3 shows the associated adjusted R2 for the proposed Mixed Model.  Similarly to the change in SE the adjusted R2 is a composite of the individual EqLat or TP results from the extended model with the exception of the results for layers 3,4, and 5 at Mauna Loa where both predictors are included concurrently.  At these layers the adjusted R2 in some cases matches the higher Adj R2 values of the two predictors, and in others improves with the combination of QBO and TP or EqLat.

| LOTUS Model Proxy Tests: (Adjusted R2 of Model) | | | | | | | | | | | | | | | | |
|---|---|---|---|---|---|---|---|---|---|---|---|---|---|---|---|---|
| Height | Umkehr | Arosa/Davos | | | OHP | | | Boulder | | | MLO | | | Lauder | | |
| (hPa) | Layer | UMK | COH | SND | UMK | COH | SND | UMK | COH | SND | UMK | COH | SND | UMK | COH | SND |
| 1-2 | 9 | | 0.42 | | | 0.37 | | | 0.36 | | | 0.11 | | | 0.32 | |
| 2-4 | 8 | 0.23 | 0.39 | | 0.14 | 0.31 | | 0.17 | 0.39 | | 0.11 | 0.32 | | 0.18 | 0.34 | |
| 4-8 | 7 | 0.35 | 0.35 | | 0.31 | 0.41 | | 0.27 | 0.33 | | 0.26 | 0.32 | | 0.17 | 0.27 | |
| 8-16 | 6 | 0.31 | 0.35 | | 0.33 | 0.45 | | 0.33 | 0.40 | | 0.40 | 0.51 | | 0.25 | 0.23 | |
| 16-32 | 5 | 0.34 | 0.38 | 0.26 | 0.25 | 0.51 | 0.21 | 0.31 | 0.40 | 0.19 | 0.40 | 0.35 | 0.37 | 0.42 | 0.41 | 0.24 |
| 32-63 | 4 | 0.21 | | 0.22 | 0.29 | | 0.32 | 0.19 | | 0.13 | 0.34 | | 0.35 | 0.42 | | 0.25 |
| 63-127 | 3 | 0.24 | | 0.23 | 0.42 | | 0.26 | 0.22 | | 0.13 | 0.14 | | 0.21 | 0.25 | | 0.19 |

**Table A3: Adjusted R2 for the Mixed Model**

**Augmented Mixed Model**

It is hard to ignore the substantial reduction of SE when adding the AO/AAO predictor especially for layers 3,4 and 5 at Mauna Loa, and for layers 3 and 4 at Arosa/Davos. The results for OHP layers 3 and 4 are still compelling, though somewhat less so.  So we explore the addition of AO/AAO at these three stations only, for the layers specified. **Table A4** summarizes the predictor choices for this Augmented Mixed Model.

| LOTUS Augmented Mixed Model | | | | | |
|---|---|---|---|---|---|
| | Arosa/Davos | OHP | Boulder | MLO | Lauder |
| **Layer** | | | | | |
| **9** | EqLat | EqLat | EqLat | Ref | EqLat |
| **8** | EqLat | EqLat | EqLat | Ref | EqLat |
| **7** | EqLat | EqLat | EqLat | Ref | EqLat |
| **6** | EqLat | EqLat | EqLat | EqLat | EqLat |
| **5** | EqLat | EqLat | EqLat | EqLat, QBO, AO/AAO | EqLat |
| **4** | TP, AO/AAO | TP, AO/AAO | TP | TP, QBO CD, AO/AAO | TP |
| **3** | TP, AO/AAO | TP, AO/AAO | TP | TP, QBO CD, AO/AAO | TP |





Table A4:  Details of additional predictor choices for each level and station in the Augmented Mixed Model . This differs from Table A1 by adding AO/AAO at some levels for Arosa/Davos, OHP and Mauna Loa.

| LOTUS Model Proxy Tests: (% Difference in Std. Error of Model) | | | | | | | | | | | | | | | | |
|---|---|---|---|---|---|---|---|---|---|---|---|---|---|---|---|---|
| Height | Umkehr | Arosa/Davos | | | OHP | | | Boulder | | | MLO | | | Lauder | | |
| (hPa) | Layer | UMK | COH | SND | UMK | COH | SND | UMK | COH | SND | UMK | COH | SND | UMK | COH | SND |
| 1-2 | 9 | | 8.35 | | | 2.74 | | | 1.94 | | | 0.00 | | | 2.85 | |
| 2-4 | 8 | -0.47 | 0.68 | | 0.09 | 1.03 | | -0.39 | 1.53 | | 0.00 | 0.00 | | 0.98 | 3.14 | |
| 4-8 | 7 | 3.75 | 3.04 | | 2.08 | 1.86 | | 5.41 | 4.08 | | 0.00 | 0.00 | | 0.53 | 1.21 | |
| 8-16 | 6 | 6.11 | 8.36 | | 2.54 | 10.88 | | 2.39 | 7.75 | | 0.55 | 0.82 | | 3.44 | 7.72 | |
| 16-32 | 5 | 7.93 | 10.72 | 5.87 | 1.92 | 13.33 | -0.56 | -1.85 | 0.00 | 0.90 | 19.39 | 13.32 | 15.70 | 0.82 | 3.91 | 0.00 |
| 32-63 | 4 | 8.71 | | 9.96 | 6.13 | | 7.04 | 3.35 | | 2.97 | 20.51 | | 10.45 | 8.03 | | 4.13 |
| 63-127 | 3 | 20.30 | | 18.49 | 13.48 | | 5.46 | 6.81 | | 2.46 | 6.00 | | 4.85 | 9.76 | | 4.41 |

Table A5:  Change in the SE of the trend using the Mixed Model.

Table A5 displays the change in the SE from the Reference Model now for the Augmented Mixed Model.  Adding AO/AAO at Arosa/Davos (layers 3 and 4) and Mauna Loa (layers 3 to 5) greatly reduces the SE beyond that of the Mixed Model results in Table A2.  For OHP (layers 3 and 4) the impact is less dramatic for Umkehr.  For sonde measurements at layer 4 the AO/AAO addition has no impact beyond the Mixed Model; for layer 3 the addition of AO/AAO results in less reduction of the SE.

| LOTUS Model Proxy Tests: (Adjusted R2 of Model) | | | | | | | | | | | | | | | | |
|---|---|---|---|---|---|---|---|---|---|---|---|---|---|---|---|---|
| Height | Umkehr | Arosa/Davos | | | OHP | | | Boulder | | | MLO | | | Lauder | | |
| (hPa) | Layer | UMK | COH | SND | UMK | COH | SND | UMK | COH | SND | UMK | COH | SND | UMK | COH | SND |
| 1-2 | 9 | | 0.42 | | | 0.37 | | | 0.36 | | | 0.11 | | | 0.32 | |
| 2-4 | 8 | 0.23 | 0.39 | | 0.14 | 0.31 | | 0.17 | 0.39 | | 0.11 | 0.32 | | 0.18 | 0.34 | |
| 4-8 | 7 | 0.35 | 0.35 | | 0.31 | 0.41 | | 0.27 | 0.33 | | 0.26 | 0.32 | | 0.17 | 0.27 | |
| 8-16 | 6 | 0.31 | 0.35 | | 0.33 | 0.45 | | 0.33 | 0.40 | | 0.40 | 0.51 | | 0.25 | 0.23 | |
| 16-32 | 5 | 0.34 | 0.38 | 0.26 | 0.25 | 0.51 | 0.21 | 0.31 | 0.40 | 0.19 | 0.44 | 0.53 | 0.40 | 0.42 | 0.41 | 0.24 |
| 32-63 | 4 | 0.23 | | 0.25 | 0.29 | | 0.34 | 0.19 | | 0.13 | 0.42 | | 0.39 | 0.42 | | 0.25 |
| 63-127 | 3 | 0.31 | | 0.31 | 0.44 | | 0.26 | 0.22 | | 0.13 | 0.19 | | 0.26 | 0.25 | | 0.19 |

Table A6: Adjusted R2 for the Augmented Mixed Model

Table A6 displays the Adj R2 for the Augmented Mixed Model. Adding AO/AAO improves the Adj R2 results for Arosa/Davos and MLO and has little to no impact at OHP.  Based on the criteria outlined at the beginning of this appendix, we assign the Augmented Mixed Model as the 'Full Model' in the body of this paper.

**Code/Data availability:** All dataset used in this study are publicly available at the website



https://gml.noaa.gov/aftp/ozwv/Publications/2023_Umkehr_Ozone_Trends_Paper/.
**Competing interests:** The authors declare that they have no conflict of interest.
**Author contributions:** IP and JW conceptualized the paper, and IP led the paper preparation. PE, KA, and JW
performed the data analysis. KM is responsible for the production of the spatial and temporally matched ground-based
and satellite ozone profile data. JW is responsible for producing COH zonally averaged and station overpass ozone
profile records. LF is responsible for the retrieval and calibration of the OMPS data. GM, PE, KM and KA are
responsible for NOAA Umkehr measurements. EMB is responsible for measurements in Arosa/Davos. RQ is
responsible for Umkehr and ozonesonde observations in Lauder, New Zealand. BJ and PC are responsible for
ozonesonde observations in Boulder and Hilo. GA is responsible for the ozonesonde observations in OHP. RVM is
responsible for HEGIFTOM ozonesonde records and data analyses. RD, SGB, DZ provided context of the LOTUS
model use and interpretation of trend analyses. All authors contributed to the writing of the paper.
**Acknowledgements:**
This study was supported in part by NOAA grant NA19NES4320002 (Cooperative Institute for Satellite Earth System
Studies - CISESS) at the University of Maryland/ESSIC and NOAA grant NA22OAR4320151, for the Cooperative
Institute for Earth System Research and Data Science (CIESRDS).  Additional funding is from NOAA Climate
Program Office's Atmospheric Chemistry, Carbon Cycle, and Climate program (AC4), grant numbers
NA19OAR4310169 (CU)/ NA19OAR4310171 (UMD). The statements, findings, conclusions, and recommendations
are those of the author(s) and do not necessarily reflect the views of NOAA or the U.S. Department of Commerce.
The authors would like to thank the NASA/GSFC Atmospheric Chemistry and Dynamics team for the SBUV/2 v8.6
profile data, Eric Beach from the NESDIS/STAR for his help with the S-NPP OMPS data, the NOAA GML
observatory team (Boulder, MLO and Fairbanks observatories), LATMOS (OHP), and NIWA (Lauder) for Umkehr
and ozonesonde data, Wolfgang Steinbrecht of the DWD for help with interpretation of the Hohenpeißenberg
ozonesonde data, the MeteoSwiss and PMOD/WRC teams (Arosa/Davos) for Dobson Umkehr data. Some data are
associated with the Network for the Detection of Atmospheric Composition Change (NDACC) and are available
through the NDACC website (www.ndacc.org).  Additional thanks are due to Susan Strahan for Equivalent Latitude
data at each station, to Kai-Lan Chang of CIRES for discussion of statistical interpretation of the thresholds, and to
Justin Alsing for development of the LOTUS code. North American Regional Reanalysis (NARR) data provided by
the NOAA PSL, Boulder, Colorado, USA, from their website at https://psl.noaa.gov.

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
