# Peer review of "Ozone trends in homogenized Umkehr, Ozonesonde, and COH"

_EGUsphere, 2024_

## Referee Comment (RC2)

[referee-annotated manuscript omitted]

---

## Author Comment (AC1)

**Authors responses:**
**The authors thank the Reviewer for taking time to provide the valuable comments and suggestions.**

**Reviewer 1 comments**
Review of "Ozone trends in homogenized Umkehr, Ozonesonde, and COH overpass records" by Irina Petropavlovskikh et al.

**Summary and General Comments:**

The authors demonstrate improvements to the LOTUS MLR ozone trends model through addition of dynamical proxies applied to ground-based (Dobson Umkehr, ozonesonde) and satellite (NOAA COH overpass and zonal) ozone datasets. Ozone trends for the lower to upper stratosphere are first presented with the "standard" Reference LOTUS model (excluding the AOD proxy) for 2000-2020, after which individual additional proxies including tropopause pressure, Eddy Heat Flux, Equivalent Latitude, etc., are added to the model to determine the best choices for a "full" model trends calculation.

The authors find that, with a few exceptions in the lower stratosphere, the trend values are mostly unchanged in the full model. However, improvements to the model adjusted R2 values and p-values indicate that the addition of proxies specifically chosen for various stations and altitudes will lead to more confidence in trend detection, as well as the possibility of detecting trends smaller in magnitude compared to a base MLR with limited proxies (i.e., as would be used with zonally averaged data).

The paper is written exceptionally well, is highly detailed, and the decision-making process for choice of additional proxies in the model and other topics are carefully explained in the text and in extensive Appendices.

I have no major concerns with this manuscript, but I do wonder if the authors explored using the Payerne ozonesonde record in addition to the Hohenpeissenberg record for the Arosa/Davos station. The Payerne record is also extremely dense, Payerne is only 50 km farther in distance from Arosa/Davos, and that is an ECC record that does not have a correction factor applied as with the Hohenpeissenberg Brewer-Mast type ozonesondes. The inclusion of Payerne sonde trends could be illuminating.

*Authors' response:*
*We considered performing trend analyses of the Payerne ozonesonde records available from the HEGIFTOM archive (which contains ECC sondes only datasets). However, the shortness of the record (homogenization was applied to the data starting in 2002 after the Brewer-Mast to ECC sondes transition) prohibits using the LOTUS multi-linear regression method on the same time range as the other datasets. Specifically, we used the independent linear trend (ILT) approach which requires data starting a few years before 1996. The record downloaded from the NDACC archive has a step change in 2002 when Payerne station switched from the Brewer-*

*Mast version of ozonesondes to the ECC sondes as long as the Dobson normalization factor is not applied on the ECC timeseries (Stuebi et al, 2008 ). The HOH dataset is however a Brewer-Mast based time series on which the Dobson normalization factor is applied. We discussed Payerne record use with the PI, coauthor of this paper, and decided that it was hard to interpret the results of the trend analysis when it is impacted by a step change in the 1996-2020 time range. Once a new record would become available we would be happy to repeat trend analyses.*
*Stuebi, R., G. Levrat, B. Hoegger, P. Viatte, J. Staehelin, and F. J. Schmidlin (2008), In-flight comparison of Brewer-Mast and electrochemical concentration cell ozonesondes, J. Geophys. Res., 113, D13302, doi:10.1029/2007JD009091.*

**Recommendation:**
I recommend publication of this paper and have only minor and technical comments below.

**Specific and Line-by-Line Comments:**

Line 117 and 118: It looks like there are some extra parentheses on these lines.

*Response: We corrected the parenthesis in the text.*

Line 127: SHADOZ is "Southern Hemisphere Additional Ozonesondes"

*Response: We corrected the text.*

Line 133: The vertical resolution of ozonesonde data is a factor of the time response of the instrument, not altitude.

*Response: We corrected the text "depending on the balloon accent velocity and the time response of the instrument".*

Table 1: If using the OHP ozonesonde data from NDACC, I am assuming you are using the homogenized "Corrected Ozone partial pressure" ozone values in those files, correct?

*Response: Thank you for making us aware of the additional column available in NDACC files for the OHP record. We selected the corrected ozone partial pressure and repeated our trend analyses. Please find updated results for the OHP ozone sonde data in the revised paper.*

Table 2: NOAA 11 appears twice in this Table.
*Response: This is correct. NOAA 11 is used in two separate time periods, first on the ascending orbit, then again when it emerges from darkness and we use the data from the descending orbit. We now make changes in the table*

Line 263: I think you are missing a "m =" here
*Response: we added m constant to the text.*

Line 362: Change "interannual" to "interannually"

*Response: we made a change to the text.*

Line 413: The OHP sonde and Umkehr trend differences look quite large for all Layers 3-5 (not just 4), although always within the 2 standard errors.

*Response: After replacing the uncorrected OHP ozonesonde data with a homogenized record, we found that trends became less positive. Therefore, their difference from the Umkehr trends became smaller (see updated Figures). The updated ozonesonde trend uncertainties were also reduced but continue to overlap with Umkehr trend uncertainties. Additional efforts are ongoing to improve OHP ozonesonde homogenization but a new version is not yet available (private communications with PI). We updated Figures, Tables and discussion in the text.*

[Figure]

Figures 3 and 5: Suggest changing "Height (hPa)" to "Pressure (hPa)"

*Response: We made the requested change.*

Line 528: I'm not sure I would say the results point to the inability of the model to detect non-zero trends. At this point we really don't know and probably cannot say that trends are non-zero here in Layer 6.

*Response: we changed the text to "Therefore, the statistical trend model cannot separate trends from zero due to unexplained high ozone variability in this layer".*

Line 606 and 794: "lower" stratospheric ozone records…

*Response: we made requested changes to the text.*

Table 13: A plot similar to Figure 7 with the adjusted R2 values, but for the p-values for the Ref and Full models could be helpful and would keep the reader from having to flip back and forth between the two Tables 6 and 13.

Response: we replaced Table 13 with Figure 7c. This is an updated Figure (10/20/2024)

[Figure]

*Figure 7 c) P Value of the Full (dashed lines) and Reference (solid lines) model. The vertical dashed lines at 0.05, 0.1, 0,33 indicate: high certainty of trend detection (below 0.05), medium (between .05 and 0.1), low (between 0.1 and 0.33), and very low certainty or no evidence of trend detection (above 0.33).*

Lines 866-869: I think it would be useful to put these questions in the intro as well (or just move them there) to very clearly motivate this study.

*Response: Thank you for the suggestion to clarify the motivation of the paper in the introduction section. We outlined motivations in lines 89-100 that should cover the questions and the goals of the paper. Therefore, we decided to keep the question in the Conclusion section. We also added the following sentence after line 100.*

*Ability of the ground-based and ozonesonde records in capturing semi-global ozone changes is evaluated by comparing trends derived from the satellite overpass and zonally averaged records*

Line 901: Change "in case of" to "for"
*Response: We made requested change to the text,*

---

## Author Comment (AC2)

**Authors responses:**
**The authors thank the Reviewer 2 for very useful comments and suggestions. We posted our responses below lines with comments and italicized the text for tracking.**

**Reviewer 2 comments**
This paper is dedicated to improving evaluation of ozone trends using homogenized Umkehr, ozonesonde and satellite overpass data.

The use of additional dynamical proxies in the LOTUS regression model is investigated in detail, and a so-called "full model" with proxies depending on altitude and location has been applied. The authors found that the use of additional proxies did not significantly change trends but reduce trend uncertainties and improve the quality of the fit.

Please find my rather minor comments on the paper in the annotated manuscript.

Page: 1,

line 30 Aros(a)

*Response: Corrected.*

Line 32: It is better to say "standard deviation" or indicate the confidence level.

*Response: We changed "two sigma" to "two times standard error uncertainty"*

Line 34: As the above comment

*Response: We changed "two sigma" to "two times standard error uncertainty"*

Lines 54-62 "I believe, this historical information can be omitted (or significantly shortened) for this paper."

*Response: we removed these lines. Therefore, we slightly changed the text at the beginning of the following paragraph to define some abbreviations that were deleted.*

*The Long-term Ozone Trends and Uncertainties in the Stratosphere (LOTUS) study was initiated in 2016 under Stratosphere-troposphere Processes And their Role in Climate (SPARC) project to reconcile the differences in defining trend uncertainties between methods outlined in the WMO Assessment (WMO, 2014) and the SPARC/IO3C/IGACO-O3/NDACC (SI2N) study (Harris et al., 2015).*

Line 150: Please explain why you did not use HEGIFTOM data for all the ozonesonde stations used in the paper.

*Response: We considered performing trend analyses of the Payerne ozonesonde records available from the HEGIFTOM archive. However, the shortness of the record (homogenization was applied to the data starting in 2002 after the BM to ECC transition) prohibits using the LOTUS multi-linear regression method on the same time range as the other datasets. Specifically, we used the independent linear trend (ILT) approach which requires data starting a few years before 1996. The full-length Lauder homogenized record was archived at the NDACC and we used it to perform trend analysis.*

Line 277: Perhaps you mean "large aerosol level"?
*Response:  No, we meant that large $SO_2$ levels from volcanic eruption can interfere with the chemical reactions in the cathode cell of the ozonesonde instrument which leads to the low (near zero) ozone levels in the recorded profile. Although, this artifact is mostly noted in the tropospheric levels of ozonesonde profiles that are launched near the active volcanoes (i.e. Hilo, Costa Rica, etc), there might be the $SO_2$ interferences in the lower stratosphere a few days after an explosive volcanic eruption(i.e. Mt. Pinatubo, Hunga, etc). We reference Yoo et al, 2022 paper for further details. We made a change to the sentence and added more information (see bolded text).*

*Large sulfur dioxide ($SO_2$) levels **reaching the lower stratosphere following major** volcanic eruptions **(i.e. El Chichon, Pinatubo or Hunga)** can impact the validity of sonde ozone detection (Yoon et al., 2022). **However, $SO_2$ is not long lived and is soon converted to sulfate aerosols that can alter observations by ozone remote sensing systems.***
*This sentence will go before the following sentence.*
*Both Umkehr and satellite ozone profiles from SBUV and OMPS are highly uncertain and/or biased because of high aerosol load during volcanic eruptions (refs in the paper).*

Line 284: Although eliminated data during periods of high volcanic activity, there were a set of moderate volcanic eruptions that can cause ozone variability.
Therefore, I would recommend using volcanic proxy in the regression model.

*Response: The LOTUS MLR ILT model used in SGB paper included the aerosol proxy. However, Umkehr and COH data are biased during the volcanic high aerosol load periods and we made a decision to exclude these periods from analyses (see paper). After we excluded the volcanic periods, we found small negative correlations between the Solar and aerosol proxies which can bias trend detection. Therefore it was decided not to include Solar and aerosol proxies simultaneously in the version of the LOTUS MLR ILT mode we used in our study. Moreover, the changes in the Umkehr and COH retrievals are small (and smoothed) for small amounts of aerosol loading and therefore we do not expect the LOTUS ILT model to be able to attribute stratospheric ozone changes for the recent aerosol variability in these records.*

*In order to test our assumptions, we repeated analyses with the trend model that includes the aerosol proxy (AOD) and did not find any significant changes in the trends or uncertainties (see results of tests below). Therefore, we decided not to include AOD in the Full model that is described in this paper. Something else to note: we did trend analyses on the record that stops in 2020, thus we could not test the impacts of the Hunga eruption on the trends.*

*Here are two tests with the full model:*

1) *Test 1: investigating the impacts of adding AOD proxy to the Full model trend. Reference results (solid line) show a "Full" model without the AOD proxy (similar to results in the paper). We compare it to the "Full" model with AOD proxy included (dashed lines). In both cases all data are used in analyses (no gaps for volcanic period, which is different from the Full model results in the paper). As you can see, no significant changes are found in trends (or SE, not shown) especially in the middle and upper stratosphere. In the lower stratosphere small differences are found in the ozonesonde trends at Boulder and Hilo (MLO panel), and in Umkehr trends at OHP. While attributing a small portion of the trends to AOD impacts on ozone variability, the differences (large at OHP and Lauder) between ozonesonde and Umkehr trends are not reduced.*

[Figure]

2) *Test 2: investigating the impact of gaps in the data. Reference results (solid line) show results of the "Full" model without the AOD proxy, the data are removed during the volcanic periods (exactly the same as in the paper). We want to test how the AOD proxy can change trends (dashed lines, "(Full)"). In this case all data are used in analyses (similar to the test 1). As you can see, again, no significant changes are found in trends, although there is a small change in ozonesonde trends (solid line) at Hohenpeißenberg (Arosa comparisons) in the lower stratosphere, reduced difference in ozonesonde trends at Boulder, and a small increase in the differences in Umkehr trends at Boulder and*

*MLO. No visible changes are found in COH trends.*

[Figure]

Line 289: Why did you end your time series/trend analysis in 2020?

*Response: we wanted to compare with results published on Sophie Godin-Beekmann et al. 2022 paper, that uses data through 2020. The paper is mostly focused on the sensitivity study of the additional proxies impacts on trends and uncertainties rather than assessing trends changes by the extension of the record and dealing with the Hunga volcanic impact on ozone trends.*

Lines 365-367: This note is better to move from bullets to the main paper text

*Response: We decided to keep it together with the bullet that discusses the EqLat proxy.*

Line 385: Please change to R^2 everywhere in the paper.

*Response: We made changes to the requested text.*

Line 412-413: Please explain this in more detail. Do you mean the proximity of solid and dashed orange lines? Then it is better to write this conclusion after the discussion below.

*Response: We decided that the text will stay in its current place. This section discusses the impact of sampling on trends and refers to figures A12 and A13 in Appendix D. Starting with text in line 415 the discussion goes back to the results shown in Figure 3. We added "Figure 3 shows" at the beginning of line 415 to clarify the change in the discussion.*

*Line 435 discusses sampling impacts on trends in the lower stratosphere. We modified text to clarify the reference Figure in the Appendix*

*Modified txt at the end of the sentence:*

*"(See Appendix D, Figure A11)"*

Line 452: deviations

*Response: We used SE (Standard Error) as error bars to show uncertainty of the trend and the overlap of error bars indicate that trends are not statistically different.*

Line 458: DU/dec?
*Response: yes, we corrected units.*

Line 459: Is "the standard error of the trend fit" the estimated uncertainty of the linear term in Eq.(1)? If yes, please indicate this. Also please provide the information how this uncertainty is evaluated.

*Response:  We change the text to:  We use the standard error of the linear (trend) term in Equation 1 to evaluate the success of the additional proxies to improve the understanding of the trend values.  The standard error is an output of the regression code, and indicates the uncertainty in the trend value.  Smaller Standard Errors indicate increased confidence in the trend result.*

Line 499 This information is presented already in Figure 4: you can just add a vertical line R2=0.3
*Response: We added the 0.3 line in the Figure (see example below).*

Line 556: SE_ref is small, then even a small absolute difference will result in a large relative difference. I would recommend using absolute difference, SE_re

*Response: We appreciate this suggestion, however we already tried this approach and it is hard to compare significance of changes without using the reference*

Line 572 (table) I suggest coloring the cells corresponding to increased R2 compared to the reference model.

*Response: We tried to add colors to the R^2 tables and it makes it harder to interpret as we already used colors in the standard error table, but colors there correspond to the different test*

*conditions. We decided not to color code the R^2 tables, but replace tables with Figures for each Extended Model test. Also, the comparisons between R^2 from the Full and Reference model are plotted in Figure 7.*
*Example of the new Figure (with dashed line at 0.3)*

[Figure]

Line 580 a table would consist of Table 7a and 8a, for Equivalent latitude, Table 7b and 8b, and so on. In other words, I suggest to split big tables 7 and 8 into smaller tables according to proxies, and put them into corresponding subsections.

*Response: Thank you for your suggestion. We decided to keep Table 7 results together in the Table, but replaced Table 8 with Figure X that shows results of both the Reference and the Full model. We hope that this adjustment will make it easier for the reader to understand the magnitude of changes and to compare results of six Extended models.*

Line 588 Please discuss also an increase of trend uncertainties in the middle and upper stratosphere.

*Response: We added the following explanation to the text.*
*The TP proxy only explains ozone variability near the tropopause because changes in both parameters are linked to the same dynamical processes (i.e. irreversible mixing). In the middle and upper stratosphere ozone variability is not linked to the processes that change TP, thus using this proxy adds the errors to the model fit.*

Table 12. The same comment on coloring cells as for Table 8.
*Response: Thank you for your suggestion. Table 8 is now replaced with Figures X showing both the reference and extended model results. We hope you agree to this change. Once the change is approved, we will change the Figures and Table numbers.*